



# Chlorophyll shading reduces zooplankton diel migration depth in a high-resolution physical biogeochemical model.

Mathieu A. Poupon[1,2], Laure Resplandy[1,3], Jessica Garwood[4], Charles Stock[5], Niki Zadeh[5], and Jessica Y. Luo[5]

[1]Department of Geosciences, Princeton University, Princeton, NJ, USA
[2]Atmospheric and Oceanic Sciences Program, Princeton University, Princeton, NJ, USA
[3]High Meadows Environmental Institute, Princeton University, Princeton, NJ, USA
[4]College of Earth, Ocean, and Atmospheric Sciences, Oregon State University, Corvallis, Oregon, USA
[5]NOAA Geophysical Fluid Dynamics Laboratory, Princeton, NJ, USA

**Correspondence:** Mathieu A. Poupon (mpoupon@princeton.edu)

**Abstract.** Zooplankton diel vertical migration (DVM) is critical to ocean ecosystem dynamics and biogeochemical cycles, by supplying food and injecting carbon to the mesopelagic ocean (200-800 m). The deeper the zooplankton migrate, the longer the carbon is sequestered away from the atmosphere and the deeper the ecosystems they feed. Sparse observations show variations in migration depths over a wide range of temporal and spatial scales. A major challenge, however, is to

understand the biological and physical mechanisms controlling this variability, which is critical to assess impacts on ecosystem and carbon dynamics. Here, we introduce a migrating zooplankton model for medium and large zooplankton that explicitly resolves diel migration trajectories and biogeochemical fluxes. This model is integrated into the MOM6-COBALTv2 ocean physical-biogeochemical model, and applied in an idealized high-resolution (9.4 km) configuration of the North Atlantic. The model skillfully reproduces observed North Atlantic migrating zooplankton biomass and DVM patterns. Evaluation of

the mechanisms controlling zooplankton migration depth reveals that chlorophyll shading reduces by 60 meters zooplankton migration depth in the subpolar gyre compared with the subtropical gyre, with pronounced seasonal variations linked to the spring bloom. Fine-scale spatial effects (<100 km) linked to eddy and frontal dynamics can either offset or reinforce the large-scale effect by up to 100 meters. This could imply that for phytoplankton-rich regions and filaments, which represent a major source of exportable carbon for migrating zooplankton, their high-chlorophyll content contributes to reducing zooplankton

migration depth and carbon sequestration time.

## 1 Introduction

The diel vertical migration (DVM) of zooplankton is the largest synchronized movement of organism on Earth, with major implications for the ocean ecosystem dynamics and biogeochemical cycles (Hays et al., 1997; Steinberg et al., 2002; Bianchi et al., 2013b). Every day at dawn, after spending the night feeding, half of the world's zooplankton biomass leaves the surface

ocean for deeper waters, where it remains before swimming back to surface waters at dusk (Klevjer et al., 2016). Migrating zooplankton shape biogeochemical cycles by consuming organic matter in the surface waters and respiring or excreting some



of it deeper down, sequestering carbon away from the atmosphere, and potentially accounting for 10-30% of the biological carbon pump (Steinberg et al., 2000; Boyd et al., 2019; Aumont et al., 2018; Archibald et al., 2019; Nowicki et al., 2022). This efficient vertical transport of organic matter by DVM influences the depth at which nutrients are regenerated (Steinberg et al.,

2002; Longhurst and Glen Harrison, 1988; Bianchi et al., 2013b, 2014), intensifies respiration and oxygen depletion at the upper margin of oxygen minimum zones (around 200-500 meters deep, Bianchi et al., 2013a; Aumont et al., 2018). DVM also contributes to structuring trophic interactions by providing energy to mesopelagic ecosystems (Kelly et al., 2019), triggering cascade migration of lower trophic levels (Bollens et al., 2011) and altering the foraging performance of visual predators such as fish and marine mammals by hiding in the dark (Benoit-Bird and Moline, 2021; Chambault et al., 2024).

Zooplankton migration depth is a key factor for understanding the impact of DVM on carbon and oxygen cycles, and ecosystem dynamics. The deeper migrating zooplankton egest, excrete or respire material from the surface, the deeper nutrients are regenerated and oxygen is consumed, and the longer the carbon they release is sequestered away from the atmosphere (Boyd et al., 2019). The fecal pellets they egest can also serve as a food source for ecosystems at that depth and below (Kelly et al., 2019). Migration depth observations vary by several hundred meters over multiple spatial and temporal scales. For example,

at fine spatial scales, Powell and Ohman (2015) have shown that zooplankton migrate 200 meters deeper on one side of a 20 km-wide front than on the other. At the basin scale, Bianchi et al. (2013a) showed that zooplankton swam about 200 meters deeper in the subtropical gyre than in the North Atlantic subpolar gyre. Temporally, Omand et al. (2021) have shown that migration depth can fluctuate up to 50 meters within a single day, while Hobbs et al. (2021) have documented the existence of a seasonal cycle in zooplankton migration depth. However, the mechanisms controlling zooplankton diel vertical migration

patterns and their biogeochemical consequences are still poorly quantified (Bandara et al., 2021).

   Light is often considered the main cue that determines zooplankton migration timing and depth (Brierley, 2014). The most common hypothesis, known as the "preferendum hypothesis", states that migrating organisms modify their depth when irradiance changes, to remain in a preferred light level, or isolume (Ewald, 1910; Michael, 1911; Russell, 1927). This behavior keeps them hidden from visual predators (Zaret and Suffern, 1976). Numerous studies supporting this hypothesis have shown that

factors modifying irradiance at the ocean surface, such as sunlight, moonlight, cloud cover, ice cover or eclipses (Strömberg et al., 2002; Haren and Compton, 2013; Last et al., 2016; Omand et al., 2021; Flores et al., 2023), or in the water column, such as water transparency and chlorophyll content could modify the migration depth (Dickson, 1972; Powell and Ohman, 2015; Hobbs et al., 2021). Migration timing can further be modified by other exogenous factors such as predation pressure (Ohman et al., 1983; Lass and Spaak, 2003; Bollens and Frost, 1989) or food availability (Pijanowska and Dawidowicz, 1987; Beklioglu

et al., 2008), or endogenous ones such as circadian rhythms (Harris, 1963). Similarly, migration depth can also be modified by environmental parameters such as low oxygen concentrations (Bianchi et al., 2013a), temperature or salinity (Kimmerer et al., 1998; Glaholt et al., 2016), and endogenous ones such as zooplankton size (Ohman and Romagnan, 2016). However, the limited number of observations makes it challenging to disentangle and quantify the individual contribution of these different factors on migration depth spatial and temporal variations.

Although numerical models can provide further insights, modeling diurnal vertical migration poses several challenges. First, migration is affected by several physical, chemical and biological variables (e.g. light, oxygen concentration, food availability





and predation exposure), requiring models that capture these complex processes, as well as a realistic representation of zooplankton physiology key to biogeochemical impacts (e.g. egestion, excretion, respiration, growth and mortality rates). Second, diel vertical migration is a rapid process, where zooplankton reach migration velocities ranging from 3 to 15 cm.s$^{-1}$ (Bianchi

and Mislan, 2016), and is modulated at various temporal and spatial scales. Modeling such fast zooplankton vertical migration over a wide period and ocean area at high spatial and temporal resolution is therefore limited by computational cost. Existing models fail at capturing this complexity. One-dimensional vertical models explicitly represent zooplankton physiology and migration trajectory based on isolumes (Bianchi et al., 2013b; Nocera et al., 2020), or taking into account exposure to food and predators to maximize fitness (Pinti et al., 2019), but do not represent ocean circulation. Conversely, three-dimensional

global biogeochemical models capturing ocean circulation represent vertical migration in parameterized form, where the biogeochemical fluxes created are distributed around a migration depth determined by an isolume or oxygen threshold (Aumont et al., 2018; Archibald et al., 2019; Nowicki et al., 2022). In this case, migration trajectories are not explicitly modeled and the seasonal cycle is not always represented. Furthermore, none of these modelling approaches capture the effect of fine-scale dynamics (< 100 km), such as fronts and eddies, on zooplankton migration patterns. To improve our understanding of the impact

of DVM on biogeochemical and ecosystem dynamics, we need to develop models explicitly representing both the migration trajectories and physiology of migrating zooplankton, as well as their interactions with the environment over a wide range of space and time scales.

In this study, we develop a migrating zooplankton module and fully integrated it into a coupled physical-biogeochemical model of the North Atlantic. This model includes a range of biological and physical mechanisms that affect zooplankton mi-

gration patterns and physiology over multiple spatial and temporal scales. We use an idealized double gyre high-resolution configuration of the North Atlantic, built using an ocean circulation model (Adcroft et al., 2019) coupled with a biogeochemical model (Stock et al., 2020). Our idealized configuration replicates the characteristic biophysical dynamics of the subtropical and subpolar gyres, their seasonal cycle and fine-scale dynamics. Our migrating zooplankton module includes two migrating zooplankton size classes (medium, large) and explicitly represents their migration trajectory, controlled by light, oxygen con-

centration, size and prey distribution. Zooplankton physiology is also explicitly modeled and depends on environmental factors (e.g. temperature) and individual factors (e.g. zooplankton feeding activity or swimming). We show that our zooplankton migration model reproduces the contrasts in biomass, timing and migration depth of migrating zooplankton observed in the North Atlantic. We identify and quantify the mechanisms responsible for modulating the migration depth of zooplankton in the North Atlantic (surface irradiance, chlorophyll shading and ocean transport). We explore their variation across seasons, biomes (sub-

tropical vs subpolar) and spatial scales (gyre scale vs fine scales), and show that chlorophyll shading is the dominant process structuring migration depth at all these scales.

## 2 Materials and Methods

We develop an idealized double gyre ocean model reproducing the North Atlantic biophysical ocean dynamics (Sect. 2.1). Our model is coupled to the biogeochemical module (COBALTv2), and extended in this study to integrate two vertically





migrating zooplankton (COBALTv2-DVM, Sect. 2.4). Our new module represents realistic migration trajectory (Sect. 2.4.1), visual predation (Sect. 2.4.2) and zooplankton physiology (Sect. 2.4.3). We also develop a framework to quantify the processes modulating zooplankton migration depth (e.g., irradiance seasonality, chlorophyll shading effect at large and fine scale, vertical transport, see Sect. 2.5).

## 2.1 Idealized double gyre experiment

### 2.1.1 Configuration


We use an idealized double gyre reproducing a biophysical dynamics characteristic of the North Atlantic Ocean, including a low-productivity subtropical gyre and a highly productive subpolar gyre separated by a jet analogous to the Gulf Stream, as well as a deep winter convection region (Fig. 1). Idealized double gyre models are useful tools to examine the influence of gyre circulation and eddies on ventilation (Marshall et al., 2002; Radko and Marshall, 2003; Henning and Vallis, 2004), and
phytoplankton and biological carbon pump dynamics (Lévy et al., 2015; Resplandy et al., 2012, 2019; Couespel et al., 2021).

Here, the double gyre model was built using the Modular Ocean Model version 6 (MOM6 Adcroft et al., 2019). The domain is a square of 3570 km side length, 4000 m depth, 9 km horizontal resolution, 2-5 m vertical resolution between 0 and 100 m depth, 10-100 m between 100 and 1000 m depth and up to 250m below. The configuration is centred around 40°N and under the $\beta$-plane approximation ($f_0 = 9.35 \times 10^{-5}$ s$^{-1}$, $\beta = 1.75 \times 10^{11}$ m$^{-1}$ s$^{-1}$). The four vertical boundaries are solid imper-
meable walls preventing any lateral exchanges. This physical model is coupled to the Carbon, Ocean Biogeochemistry and Lower Trophics biogeochemical module version 2 (COBALTv2 Stock et al., 2020). COBALTv2 includes 33 biogeochemical tracers, such as the main nutrients (e.g. nitrogen, phosphorus, iron, silica) and a planktonic ecosystem. The original version of COBALTv2 included 3 phytoplankton and 3 non-migrating zooplankton, to which we added 2 migrating zooplankton (see Sect. 2.4). Vertical movement was simulated with an implicit solution of the advection-diffusion equation generalized to handle
upward and downward swimming. The resulting tridiagonal matrix was solved using the algorithm described in Press (2007). Each year, the double gyre is forced by the same idealized atmospheric zonally uniform climatology detailed below. Further details on the model parameters are in the supplementary material (Table S1).

### 2.1.2 Forcings

The ocean surface is forced with an idealized wind field (Fig. A1a). The zonal wind stress follows a sinusoidal profile, minimal
at the northern and southern boundaries of the domain ($-0.1$ N m$^{-2}$), maximal at 40° N (0.1 N m$^{-2}$) and varying by 10% in amplitude according to a seasonal cycle (e.g., higher values in december). The meridional wind stress is zero. The radiative forcing is derived from the ERA5 atmospheric reanalysis (Hersbach et al., 2020) averaged over the 1979-2022 period (Fig. A1g). The temporal resolution of ERA5 (1h) accurately resolves the daily cycle of irradiance required for vertical migration modeling. The forcing is calculated as the zonal mean over a thin band of 2° longitude located in the middle of the North
Atlantic (longitude = -35° E to -33° E, latitude = 20° N to 60° N), which avoids the daily light cycle being skewed by the time difference between the east and west Atlantic basin. The other physical atmospheric forcings (pressure, temperature,





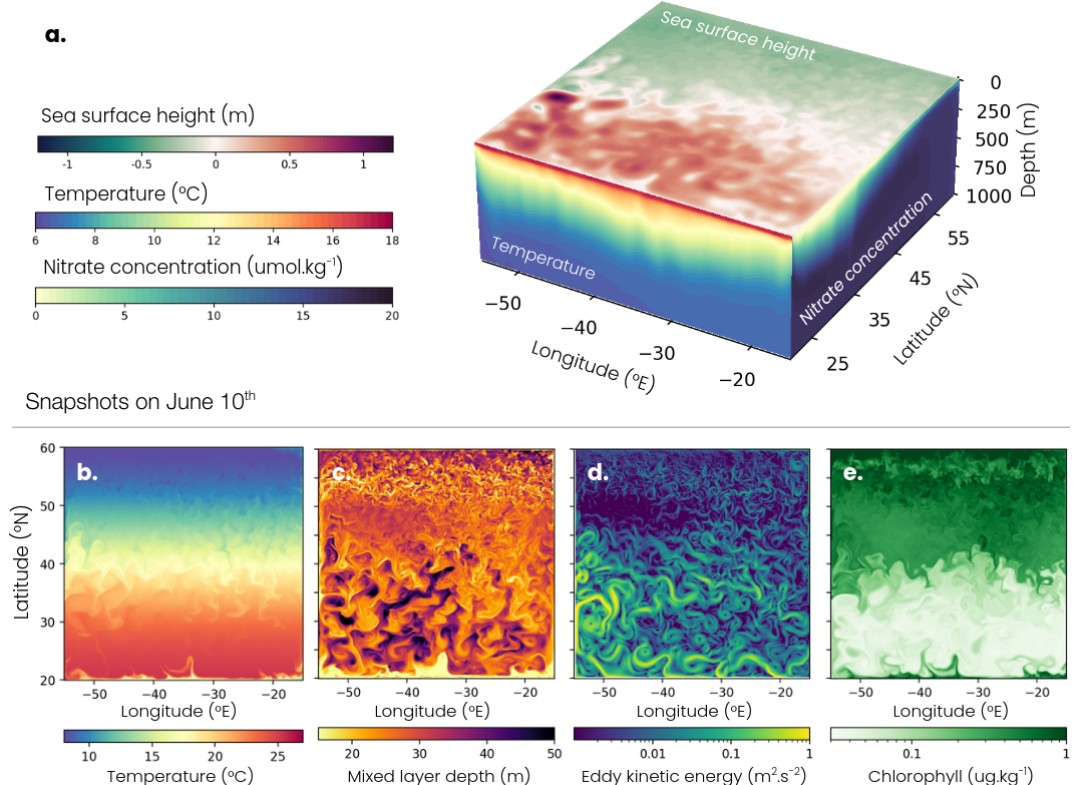

**Figure 1.** Double gyre set-up: June $10^{th}$ snapshot of (a) sea surface height (top) and depth sections of temperature at 20° N (front) and nitrate concentration at -15° E (right); (b) sea surface temperature, (c) mixed layer depth, (d) eddy kinetic energy and (e) surface chlorophyll concentration.

precipitation, humidity) are derived from the JRA55-do atmospheric reanalysis (Tsujino et al., 2018) with a 3-h time resolution averaged over the period 1958-2020 (Fig. A1b-f). They are calculated as the zonal mean over a wider region covering most of the North Atlantic Ocean (longitude = -60° E to -20° E, latitude = 20° N to 60° N). Precipitation from JRA55-do is adjusted to
ensure that the same mass of precipitation is received by the double gyre as by the North Atlantic Ocean and a mass restoration flux is applied at the surface to ensure its conservation (Fig. A1e). Iron and lithogenic material deposition (Fig. A1h-i) are calculated as the zonal mean of the last 30 years of a GFDL-ESM4 pre-industrial control simulation spanning from 1850 to 2014 (Dunne et al., 2020), over the same band as the physical forcings.

### 2.1.3 Initial state and spin-up

Temperature and salinity as well as nitrate, oxygen, phosphate, silica concentrations are initialized uniformly zonally using the World Ocean Atlas 2018 zonal mean (Boyer et al., 2018) over the region delimited by -60° E to -20° E and 20° N to 60° N. Similarly, alkalinity and dissolved inorganic carbon are initialized from GLODAPv2 (Olsen et al., 2020) and the rest of





the biogeochemical tracers are initialized from the GFDL-ESM4 simulation described above, all using zonally averaged fields
in the same region. The model is first spun-up for 1000 years at coarse resolution (85 km) and then run for 25 years at a
eddy-permitting resolution (9 km, Table S1). This paper focuses on the last 5 years at 9 km resolution

## 2.2   Observations

To evaluate the biophysical dynamics simulated in the model, we use multiple observational data products. The nutrient con-
centration, temperature and salinity used are taken from the World Ocean Atlas 2018 (Garcia et al., 2019, Fig. A2), the mixed
layer depth from a compilation of hydrographic sections between 1941 and 2002 from de Boyer Montégut et al. (2004), sur-
face chlorophyll concentration from a combination of satellite observations between 1997 and 2020 from Sathyendranath et al.
(2019) and net primary productivity is derived from Moderate-Resolution Imaging Spectroradiometer (MODIS) measurements
between 2002 and 2023 processed by three algorithms (Eppley, CbPM and CAFE Behrenfeld and Falkowski, 1997; Westberry
et al., 2008; Silsbe et al., 2016).

   The biomass of migrating zooplankton is evaluated using estimates derived from LIDAR observations between 2007 and
2019 (Behrenfeld et al., 2019) and net samples at the Bermuda Atlantic Time Series (BATS) station between 1994 and 2010
(Steinberg et al., 2012). We also use estimates of migration depth and time spent at depth by migrating zooplankton derived
from acoustic doppler current profiler (ADCP) measurements taken between 1990 and 2010 and compiled by Bianchi and
Mislan (2016).

## 2.3   Model biome definition and biogeochemical evaluation

Our model reproduces the characteristic physical and biogeochemical dynamics of four biomes of the North Atlantic: a low-
productivity subtropical gyre, a productive subpolar gyre, an energetic jet in between with intense mesoscale eddy activity, and
a deep winter convection region in the northern part of the domain (Fig. 1, Fig. 2 and Fig. A2). The subtropical, subpolar and
jet biomes are defined by thresholds of annual surface chlorophyll concentration (0.15 and 0.35 mg m$^{-3}$), as used in previous
studies (Resplandy et al., 2012, 2019). The convecting region is identified as a subregion of the subpolar region, using the
200-meter threshold on the annual mean mixed layer depth. Our main conclusions are not sensitive to these thresholds. Note
that the wind creates an upwelling along the wall at the southern boundary and stimulates ocean productivity with annual
mean chlorophyll concentrations higher than 0.35 mg m$^{-3}$ (Fig. 2d). This upwelling, representative of the equatorial dynamics
occurring further south in nature, is excluded from the analysis using the chlorophyll thresholds given above.

   The mean biophysical conditions and seasonal cycle simulated within the biomes are similar to those observed in the ocean.
In both observations and model, the subtropical anticyclonic gyre (elevated sea surface height, Fig. 1a) is oligotrophic at the
surface, with nitrogen concentrations below 1 $\mu$molN kg$^{-1}$, and phosphorus concentrations below 0.1 $\mu$molP kg$^{-1}$ in the
first 100 meters (Fig. A2a-d). The model also reproduces the observed stratification throughout the year, with a mixed layer
varying between 30 and 60 m seasonally (Fig. 2a-c), strongly limiting the supply of nutrients to the surface. Consequently, net
primary productivity in the subtropical gyre is low (between 170 and 250 mgC m$^{-2}$ d$^{-1}$ on average in model and observation-
based estimates) and varies little seasonally (Fig. 2d-f). In contrast, the subpolar cyclonic gyre (depressed sea surface height)







**Figure 2.** Map of annual mean (a,b) mixed layer depth (MLD), (d,e) surface chlorophyll concentration and (h,i) migrating zooplankton biomass in (a,d,h) the double gyre model and (b,e,i) observation-based data products. The subtropical, jet and subpolar biomes are delimited by solid black lines (mean surface chlorophyll concentration = 0.15 and 0.35 $mg\ m^{-3}$), while the subpolar and convective biomes are delimited by a dashed line (mean MLD = 200 m, see Methods). Mean seasonal cycle of (c) MLD, (f) net primary productivity and (j) migrating zooplankton biomass in the subtropical biome (orange) and the subpolar biome (blue) in the the double gyre model (solid lines) and observation-based products (dashed lines). Shading and error bars correspond to two standard deviations around the average. Orange star in (i) indicate the Bermuda Atlantic Time-series Study (BATS). See Sect. 2.2 for observational product details.

is a nutrient-rich environment. Nitrate concentrations range from 8 to 17 $\mu$molN kg$^{-1}$ in the top 100 meters, and phosphate concentrations from 0.3 to 0.8 $\mu$molP kg$^{-1}$ (Fig. 1a, Fig. A2a-d) in both observations and model. This is partly due to high seasonality, with a modeled mixed layer averaging 140 m in winter (125 m in observations, Fig. 2c), bringing nutrients to the





surface. Consequently, net primary productivity in the subpolar gyre is high and seasonally marked (Fig. 2f). It increases from
January onwards, peaks in May (920 mgC m$^{-2}$ d$^{-1}$ in model, 650-930 mgC m$^{-2}$ d$^{-1}$ in observations) and falls back below
500 mgC m$^{-2}$ d$^{-1}$ at the end of summer.

Finally, at the northern boundary, the wind creates a downwelling and a deep convecting water column characterized by
winter mixed layers exceeding 1000 meters (Fig. 2a). These deep convection events can be considered as representative of
ocean dynamics occurring further north in the Atlantic Ocean, such as the Irminger and the Labrador Seas (Fig. 2b). These
convection events bring a high concentration of nutrients to the surface (>12 $\mu$molN kg$^{-1}$ and > 0.7 $\mu$molP kg$^{-1}$, Fig. 1a,
Fig. A2a-d) and sustain a biome productivity of 420 mgC m$^{-2}$ d$^{-1}$ on annual average, reaching up to 1250 mgC m$^{-2}$ d$^{-1}$ in
spring.

These large-scale dynamics are locally modulated by fine-scale structures such as eddies and fronts (Fig. 1b-e). The jet
hosts the most energetic fine-scale dynamics (Fig. 1d, Fig. A3), because it is unstable and therefore induces eddy formation.
Many eddies are present in the subtropical gyre, since the horizontal resolution of the model is sufficient to resolve mesoscale
dynamics (>100 km). Conversely, in the subpolar gyre, the resolution is not sufficient to fully resolve these eddies, leading to
less coherent and more filamentary turbulent structures. As we will show in this study, these features introduce variability in
the gyre and seasonal dynamics and are key to understand zooplankton migration depth variability.

### 2.4 Migrating zooplankton model

We developed the COBALTv2-DVM module by adding two migrating zooplankton to the pre-existing COBALTv2 biogeo-
chemical module (Stock et al., 2020). These migrating zooplankton are the same as the medium (0.2-2 mm) and large (2-20
mm) zooplankton of COBALTv2 in terms of prey preferences and density-dependent higher trophic level predator losses.
However, they perform diel vertical migration (2.4.1), enabling them to escape visual predation (2.4.2). They also have a
compartmentalized physiology that temporally decouples ingestion, digestion, respiration and growth (2.4.3), to realistically
represent the matter fluxes which differ when zooplankton feed at the surface and when they dive, as highlighted by the work
of Bianchi et al. (2013b).

### 2.4.1 Migration model

The migration trajectory of zooplankton is controlled by the light intensity, oxygen concentration and vertical distribution of
its prey (Fig. 3a). Migration is assumed to be solely vertical and is controlled by the vertical migration velocity ($w$).
During the day, migrating zooplankton dive following an isolume, i.e. a preferred light intensity level (I$_{iso}$, medium: I$_{iso}$ =
10$^{-3}$ W m$^{-2}$, large: I$_{iso}$= 10$^{-4}$ W m$^{-2}$). As a result, large zooplankton dive deeper than medium zooplankton. The migration
speed $w(z)$ is close to the maximum swimming speed ($w_0$) away from the isolume and starts to decrease, according to a Monod
equation, when it approaches about 50 meters from its isolume:

$$w(z) = w_0 \frac{|\frac{1}{K_{Blue}} \ln(\frac{I_{iso}}{I(z)})|}{K_{dz} + |\frac{1}{K_{Blue}} \ln(\frac{I_{iso}}{I(z)})|} \tag{1}$$





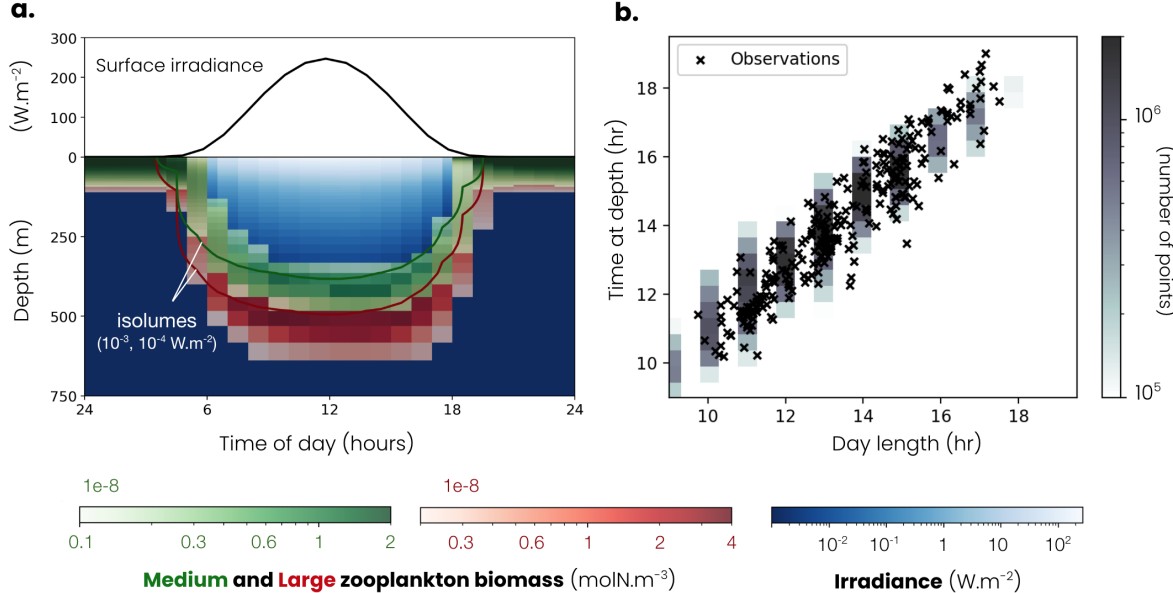

**Figure 3.** (a) Top of panel - Mean daily surface irradiance cycle (black line). Bottom of panel - Mean daily cycle of migrating zooplankton biomass (medium: green color fill, large: red color fill), migrating zooplankton isolume depth (medium: green line, isolume = $10^{-3}$ W m$^{-2}$. large: red line, isolume = $10^{-4}$ W m$^{-2}$) and irradiance (blue color fill). (b) Time spent at depth by migrating zooplankton as a function of day length in observations (cross) and model (distribution of model points shown by grey color fill).

With $K_{blue}$ = 0.0232 m$^{-1}$ the light attenuation coefficient for optically pure seawater (Manizza et al., 2005), $K_{dz}$ = 50 m the half-saturation distance, I(z) the irradiance at depth $z$, and $w_0$ the maximum migration velocity determined from observations (medium $w_0$ = 6 cm s$^{-1}$, large: $w_0$ = 8 cm s$^{-1}$, Fig. A7). The model also includes a migration depth limitation by an oxygen threshold, set to 10 $\mu$molO$_2$ kg$^{-1}$, similarly to Aumont et al. (2018). However, this limitation is not utilized in our simulation as oxygen concentrations are higher than 10 $\mu$molO$_2$ kg$^{-1}$, and therefore do not limit migration. The sensitivity of the results to the migration model parameters ($w_0$, I$_{iso}$, K$_{dz}$) is assessed by varying their values by ± 50% (see Fig. B3).

At night, migrating zooplankton return to the surface to feed. To reproduce foraging behavior, migrating zooplankton distributions are redistributed according to the distribution of its preys, using:

$$ZooExcess(z) = \int\limits_0^z \left( \frac{Z(z')}{\overline{Z}} - \frac{P(z')}{\overline{P}} \right) dz' \qquad (2)$$

Where $Z(z')$ and $P(z')$ are the concentrations of zooplankton and its prey at depth $z'$. $\overline{Z}$ and $\overline{P}$ is their mean concentration over the entire water column. At a given depth $z_0$, if $ZooExcess(z) > 0$, the zooplankton is in relative excess with respect to its prey between the surface and this depth. Therefore, zooplankton dive ($w(z) = w_0$) to restore the balance. If $ZooExcess(z) < 0$, the zooplankton is in relative deficit to its prey. Therefore, zooplankton rise to fill this imbalance ($w(z) = -w_0$).




### 2.4.2 Visual predation model

In the ocean, zooplankton migrate vertically to escape visual hunting predators. To represent this competitive advantage of migrating zooplankton over non-migrating, we implemented the visual predation model derived by Bianchi et al. (2013b) in COBALTv2-DVM. In this model, a fraction of the predation rate of top predators ($I_{HP}$) on zooplankton (Z), corresponding to visual predation, increases with irradiance (I(z), Fig. A8):

$$I_{HP} = I_{HP}^{max} \cdot \frac{Z}{K_Z + Z} \cdot \left( (1-\alpha) + \alpha \frac{I(z)}{(\frac{K_Z}{K_Z+Z}) \cdot K_{irr} + I(z)} \right) \tag{3}$$

Where $I_{HP}^{max}$ is the maximum predation rate of top predators, which varies with temperature and oxygen concentration, $K_Z$ = 1.25 $\mu$molN kg$^{-1}$ is the half saturation constant for prey density dependence, $\alpha$ = 0.9 is the maximum predation fraction due to visual predators and $K_{irr} = 10^{-1}$ W m$^{-2}$ is the half saturation response for light limitation. At the surface during the day, irradiance is sufficient (I(z)$\gg$ $K_{irr}$) and does not limit visual predation: $I_{HP}(z) \approx I_{HP}^{max} \frac{Z}{K_Z+Z}$. During the night or at depth, the absence of light ($I(z) \ll K_{irr}$) inhibits visual predation and limits the predation rate to 10% of its maximum value: $I_{HP}(z) \approx (1-\alpha)I_{HP}^{max} \frac{Z}{K_Z+Z}$.

### 2.4.3 Zooplankton physiological model

To reproduce realistic biogeochemical fluxes produced by zooplankton digestion, respiration, and death, we developed a six-compartment model. The model, inspired by that of Bianchi et al. (2013b), was extended to represent phosphate, iron and silica dynamics as well as the thermal dependence of the gut clearing rate. The model represents nitrogen ($N^{gut}$), phosphorus ($P^{gut}$), iron ($Fe^{gut}$) and silica ($Si^{gut}$) in the zooplankton gut, as well as nitrogen metabolites ($N^{metab}$) and nitrogen body biomass ($N^{body}$, Fig. 4).

First, zooplankton fill their guts with the nitrogen, phosphorus, iron and silica contained in their prey. The gut content ($X^{gut} = N^{gut}$, $P^{gut}$, $Fe^{gut}$ or $Si^{gut}$) is cleared by egestion and assimilation following an e-folding timescale ranging from 15 minutes to 3 hours, depending on water temperature (Fig. A6).

$$\frac{\partial X^{gut}}{\partial t} = ingestion_X - k^{clear} \cdot X^{gut} \tag{4}$$

$$k^{clear} = k_0^{clear} + k_T^{clear} \cdot (T - T_0) \tag{5}$$

Where $T_0$ = 0 °C, and the coefficients $k_0^{clear}$ = 8 d$^{-1}$ , $k_T^{clear}$ = 4.32 °C$^{-1}$ d$^{-1}$ are derived from a linear regression on a set of digestion rate observations for different temperatures (Irigoien, 1998). The ingestion rate ($ingestion_X$) depends on prey and zooplankton abundance, and increases exponentially with temperature (see Stock et al., 2014).

Following the same ratios as the non-migrating zooplankton, 30% of the nitrogen and phosphorus consumed are egested as organic matter (medium: 70% of the egestion as particulate sinking detritus and 30% in dissolved form, large: 100% of the egestion is detritus). The remaining 70% of the consumption are assimilated ($assimilation_{N|P}$) as metabolites at a ratio of





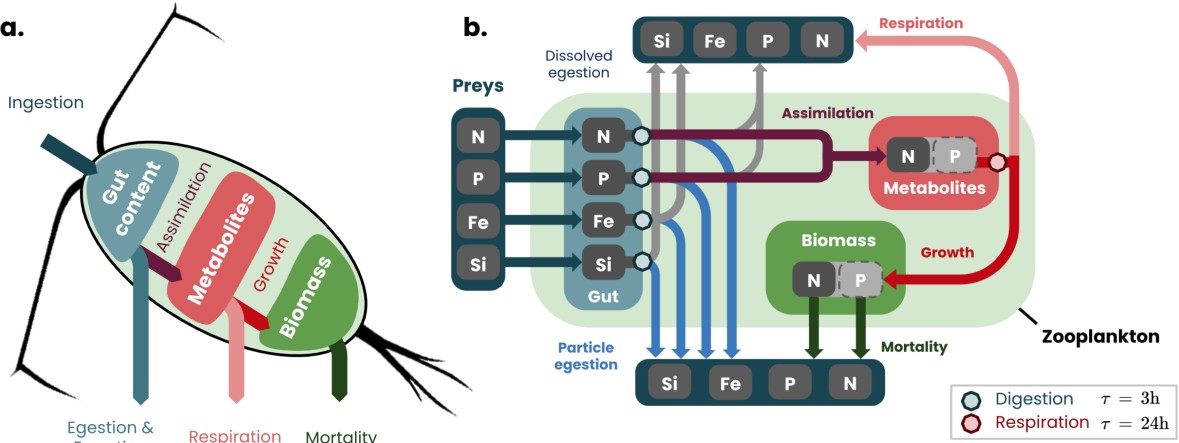

**Figure 4.** Migrating zooplankton model COBALTv2-DVM: (a) simplified and (b) detailed schemes of biogeochemical fluxes and stocks. The gut content (blue box) is filled by ingestion and cleared by egestion, excretion and assimilation, occurring between 10 min and 3 hours after ingestion. The metabolite pool (light red box) is filled through assimilation and used for zooplankton growth and respiration, occurring throughout the day. The biomass (green box) accumulates through growth and declines through mortality.

$N : P = 1 : 16$, allowing the metabolic content to be modeled with a single tracer ($N^{metab}$). Nitrogen and phosphorus in excess compared to this ratio is egested as dissolved inorganic form ($NH_4$ and $PO_4$, respectively). Silica and iron are totally egested, 30% as organic matter (medium: 70% as particulate sinking detritus and 30% in dissolved form, large: 100% as detritus) and the rest in as dissolved inorganic matter. These ratios are chosen to be the same as those used by Stock et al. (2014, 2020) for non-migrating zooplankton.

Metabolites are consumed by metabolic reactions ($metabolism$), i.e. the sum of anabolic ($growth$, through organic biomass synthesis) and catabolic ($respiration$, production of energy necessary for the functioning of the organism) reactions according to an e-folding timescale of 24h ($k^{resp} = 1$ d$^{-1}$):

$$\frac{\partial N^{metab}}{\partial t} = assimilation_{N|P} - metabolism \tag{6}$$

$$metabolism = growth + respiration = k^{resp} \cdot N^{metab} \tag{7}$$

The $respiration$ flux is the sum of three terms. The first one is the respiration of metabolites to maintain the vital functions ($respiration_{basal}$), the second one covers the energetic needs of assimilation and digestion ($respiration_{feeding}$) and the last one is linked to swimming ($respiration_{swim}$):





$$respiration = respiration_{basal} + respiration_{feeding} + respiration_{swim} \tag{8}$$

$$respiration_{basal} = \mu_{basal} \cdot N^{body} \tag{9}$$

$$respiration_{feeding} = \gamma^{resp} \cdot k^{resp} \cdot N^{metab} \tag{10}$$

$$respiration_{swim} = \frac{w}{w^{ref}} \cdot \mu_{basal} \cdot N^{body} \tag{11}$$

Where $\mu_{basal}$ is the basal respiration rate of zooplankton, which increases exponentially with temperature (see Stock et al., 2014), $\gamma^{resp}$ is the fraction of metabolic flux used to cover assimilation and digestion processes, $w$ is the migration velocity of zooplankton, and $w^{ref}$ is the reference velocity for adjusting the energy cost of swimming. The linear relationship between respiration ($respiration_{swim}$) and swimming velocity was proposed by Torres and Childress (1983) and previously used in the model developed by Bianchi et al. (2013b). Biomass production by anabolism ($growth$) is equal to the difference between the metabolic ($metabolism$) and catabolic fluxes ($respiration$). If this difference is positive, the biomass increases. If it is negative, part of the zooplankton does not have the metabolic resources to ensure its catabolic needs. This part of the zooplankton dies and its biomass is transformed into sinking detritus.

Most of the model parameters are directly derived from the literature (such as digestion rate - $k_0^{clear}$ and its temperature sensitivity - $k_T^{clear}$) or were already part of the pre-existing COBALTv2 model. Four parameters (basal respiration rate - $\mu_{basal}$, grazing rate, reference swimming speed - $w^{ref}$, maximum predation fraction due to visual predators - $\alpha$) were adjusted using 14 sensitivity experiments (see Fig. A5). We selected the parameter values to match the observed fraction and biomass of migrating zooplankton (see Fig. 2h, Fig. A5).

## 2.5 Migration depth analysis framework

In this section, we present a framework to investigate which processes control zooplankton migration depth. First, we isolate the effects of surface irradiance, chlorophyll shading, and ocean transport (2.5.1). Then, we separate their contributions according to different spatial and temporal scales (2.5.2).

### 2.5.1 Migration depth decomposition

We decompose the migration depth ($z_{mig}$) as the sum of three terms (Fig. 5) controlled respectively by irradiance in transparent water ($z_{iso,tw}$), variations in chlorophyll shading ($\Delta z_{chl}$) and variations in ocean circulation vertical transport ($\Delta z_{circ}$):

$$z_{mig} = z_{iso,tw} + \Delta z_{chl} + \Delta z_{circ} \tag{12}$$

$z_{iso,tw}$ is the isolume depth in transparent water. It represents the depth at which the isolume would be if only the effects of light absorption by water molecules were taken into account. In our model, the coefficient of light attenuation by water are constants ($k_{red} = 0.225$ m$^{-1}$ and $k_{blue/green} = 0.0232$ m$^{-1}$, (Manizza et al., 2005), so $z_{iso,tw}$ depends only on the ocean surface irradiance.





$\Delta z_{chl}$ is the chlorophyll shading effect. It is calculated as the difference between the modeled depth of the isolume and its

depth in transparent water ($\Delta z_{chl} = z_{iso} - z_{iso,tw}$). $\Delta z_{chl}$ quantifies the reduction in isolume depth due to light absorption by chlorophyll.

$\Delta z_{circ}$ is the ocean circulation effect. It is calculated as the difference between the migration depth of zooplankton and its isolume depth at zenith, i.e. when the isolume is the deepest ($\Delta z_{circ} = z_{mig} - z_{iso}$). $\Delta z_{circ}$ quantifies variations in migration depth caused by vertical physical transport (advection and mixing), which can move zooplankton away from their isolume

faster than zooplankton can respond by swimming.

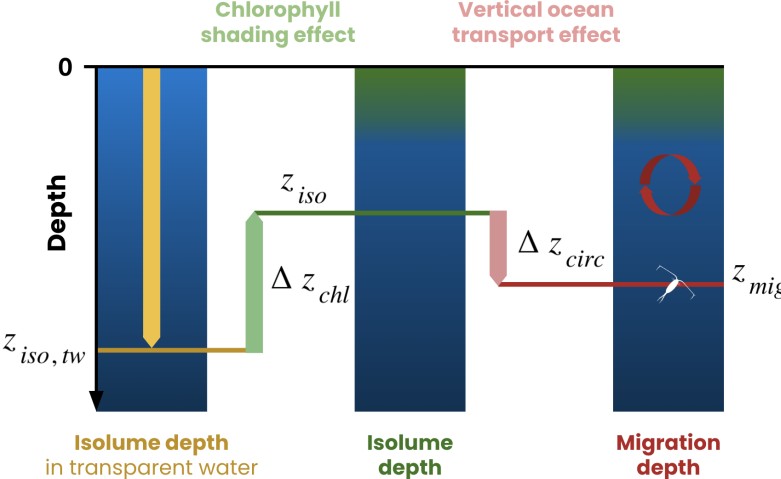

**Figure 5.** Framework used to identify the processes controlling zooplankton migration depth. The migration depth ($z_{mig}$, red line) is the sum of the isolume depth in transparent water ($z_{iso,tw}$, yellow line), the chlorophyll shading effect ($\Delta z_{chl}$, green arrow) and the vertical ocean transport effect ($\Delta z_{circ}$, pink arrow). $z_{iso}$ is the isolume depth targeted by migrating zooplankton. See Methods (Sect. 2.5) for details.

### 2.5.2 Separation of spatial scales

The chlorophyll content and vertical transport, which control migration depth, respond to large-scale dynamics (>100 km) but are also modulated by fine-scale dynamics (<100 km) such as mesoscale eddies or submesoscale fronts. To isolate these two scales of variability, we perform a spatial Reynolds decomposition. We spatially filter $\Delta z_{chl}$ and $\Delta z_{circ}$ using a Gaussian

kernel with a standard deviation of 100 km, and write them as the sum of a large-scale average ($\langle \Delta z_{chl} \rangle, \langle \Delta z_{circ} \rangle$) and a fine-scale anomaly ($z'_{chl}, z'_{circ}$) relative to this average:

$$\Delta z_{chl} = \langle \Delta z_{chl} \rangle + \Delta z'_{chl} \tag{13}$$

$$\Delta z_{circ} = \langle \Delta z_{circ} \rangle + \Delta z'_{circ} \tag{14}$$





In contrast, the isolume depth in transparent water ($z_{iso,tw}$), which depends on surface irradiance, does not vary on fine
spatial scales in our simulation because the irradiance forcing is zonally uniform. Consequently, the fine-scale term is almost
zero:

$$z_{iso,tw} = \langle z_{iso,tw} \rangle + z'_{iso,tw} \approx \langle z_{iso,tw} \rangle \tag{15}$$

To summarize, this framework provides a way to disentangle the complexity of the mechanisms modulating the migration
depth:

$$z_{mig} = z_{iso,tw} + \langle \Delta z_{chl} \rangle + \Delta z'_{chl} + \langle \Delta z_{circ} \rangle + \Delta z'_{circ} \tag{16}$$

We isolate irradiance contribution and distinguish the spatial contributions of chlorophyll shading and vertical transport at
large and fine scales. This framework does not include the limitation of migration depth by oxygen, as its concentration is high
and does not limit vertical migration in our simulation.

## 3   Results

### 3.1   North Atlantic migrating zooplankton dynamics

The model reproduces the regional and latitudinal contrasts in biomass and fraction of migrating zooplankton observed in the
North Atlantic (Fig. 2h-j, Fig. A5). In the subpolar biome, the migrating zooplankton biomass and seasonal variations are high
in both the model and LIDAR estimates ($1 \pm 0.5$ g m$^{-2}$, Fig. 2c), reflecting the high productivity and seasonality of this biome
(Fig. 2d-f). Migrating biomass is minimal in late winter ($0.69 \pm 0.34$ g m$^{-2}$), accumulates in spring and peaks in May ($1.17$
$\pm 0.57$ g m$^{-2}$). Biomass remains high throughout summer and fall ($1.11 \pm 0.54$ g m$^{-2}$), before declining in November. In
contrast, the subtropical biome is characterized by lower biomass and low seasonality in both model and observations ($0.18 \pm$
$0.12$ g m$^{-2}$, Fig. 2h-j), reflecting the low productivity and seasonality of the region (Fig. 2d-f). Migrating zooplankton account
for about half of the medium- and large-sized zooplankton in the model in both biomes, which is in line with observations from
the COPEPOD database compiled by (Aumont et al., 2018, Fig. A5).

The model also reproduces the timing of zooplankton migration. The ADCP observations compiled by Bianchi and Mislan
(2016) show a linear relationship between the length of the day and the time spent at depth by migrating zooplankton (Fig. 3b).
The longer the days, the longer migrating zooplankton remain at depth. This relationship is reproduced in the model. Migrating
zooplankton spend between 9 and 18 hours at depth depending on the length of the day. Since the day length is less variable
over the year at low latitudes than at high latitudes, the same applies to the time spent at depth by migrating zooplankton. In
the subpolar biome, migrating zooplankton stay at depth on average 10 hours in winter and 18 hours in summer, while in the
subtropical biome, it stays at depth on average 11 hours in winter and 15 hours in summer.

Finally, the model reproduces the contrasts in migration depths observed in the North Atlantic. Zooplankton migration depths
vary both across and within biomes. ADCP observations show that zooplankton dive about 100 m deeper in the subtropical
biome, where the average migration depth is 480 m, than in the subpolar biome, where it is confined to 370 m (Fig. 6a,b).



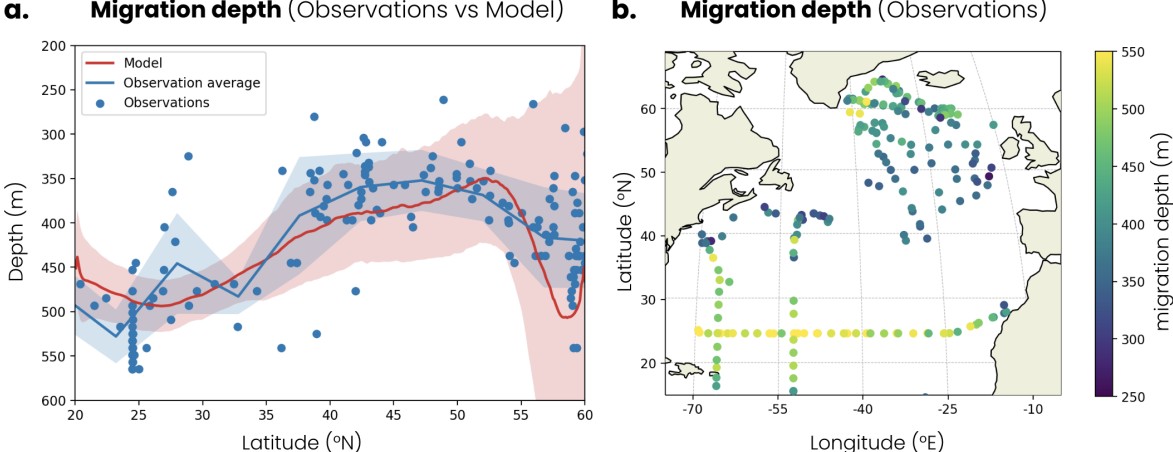

**Figure 6.** (a) Latitudinal distribution of migration depth. Blue dots are observations derived from acoustic doppler current profiler (ADCP) observations and the blue line their average per 5° latitude bins. Red line is the zonal and temporal average of the model. The shading corresponds to two standard deviations around the average. (b) Migration depth derived from ADCP observation in the North Atlantic used in panel a.

The model reproduces these large-scale contrasts with an average migration depth of 480 m in the subtropical biome and 380 m in the subpolar biome (Fig. 6a). The jet region, around 40° N, shows intermediate migration depths of around 340 m in both observations and our model. Observations also show migration depth variability within biomes, up to 100 to 200 meters around the biome average, which is relatively well captured by the model (shading on Fig. 6a). The model simulates migration depth and variability in the convective biome that are higher than observed at 55-60° N. The simulated mean migration depth

is over 450 m, whereas observations show depths of around 420 m, and variability in the model is three times greater than in observations. However, these deep convection events occur further north in nature, as in the Irminger Sea, where migrations have been observed below 500 m (Fig. 6b). Furthermore, in these deep convection regions, some zooplankton replace their daily migratory behaviour with seasonal migratory behaviour for part of the year (see discussion Sect. 4).

### 3.2 Controls of zooplankton migration depth

The observed migration depth and its variability result from the combination of processes varying on different space and time scales, which cannot be disentangled from the limited number of observations available. In the following, we therefore use the model, which reproduces the observed patterns, to investigate the mechanisms controlling the large spatial contrast between the subtropical and subpolar gyres, the variability introduced at fine-scales (> 100 km) by eddies and filaments, as well as the influence of seasonality. We identify chlorophyll shading as a major mechanism in regulating migration depth, although the

contribution of ocean transport in the convective region and the seasonality of surface irradiance are significant. Only results





for medium migrating zooplankton are presented. Results for large migrating zooplankton are similar, but where differences arise, they are indicated as such and presented in supplementary materials (Fig. B1, B2).

### 3.2.1 Mechanisms modulating zooplankton migration depth

Zooplankton migration depth is modulated by three mechanisms: surface irradiance, chlorophyll shading and ocean vertical
transport (see Methods, Sect. 2.5). We illustrate the influence of these three effects using model snapshots and a depth-latitude section across the different biomes early in the Spring bloom (March 1st, Fig. 7). First, migrating zooplankton target a specific isolume whose depth depends on surface irradiance intensity. As surface irradiance increases, light penetrates further into the water column, resulting in a deeper isolume. We track the isolume depth as if the water were transparent ($z_{iso,tw}$) to isolate this irradiance effect. For example, Figure 7 shows that on March 1st, $z_{iso,tw}$ is slightly deeper in the subtropical gyre than in
the subpolar gyre, because surface irradiance at zenith is stronger at low latitude than at high latitude (yellow line and arrow on Fig. 7d).

Seawater contains chlorophyll, which modifies light penetration in the water column by changing its opacity. As chlorophyll content increases, water opacity and chlorophyll shading increase resulting in a shallower isolume ($z_{iso}$, Fig. 7, green line on panel d). Shallow migration regions are characterized by high chlorophyll content and deep migration regions by low
chlorophyll content. This chlorophyll shading effect ($\Delta z_{chl}$, green arrow), acts on large scale, between the subtropical and subpolar biomes, but also at the scale of eddies and filaments, where migration depths can vary by up to 200 m between points that are only 20 km apart with chlorophyll content that differ by more than 60 mg m$^{-2}$ (Fig. 7a,b).

Finally, migrating zooplankton can be transported away from their isolume when ocean vertical transport associated with advection and/or turbulent mixing is high. For example in the convective biome, vertical ocean transport ($\Delta z_{circ}$, red arrow)
carries migrating zooplankton more than 200 m below their isolume (Fig. 7d). This occurs during deep convection events where the mixed layer exceeds 1000 m (Fig. 7c).

### 3.2.2 Controls of migration depth spatial variability at large-scale

We generalize our analysis by evaluating the average annual effect of irradiance, chlorophyll shading and vertical transport on migration depth across the four biomes (Fig. 8). We find that the spatial contrast in migration depth between the different
biomes is mainly due to differences in chlorophyll shading. On annual average, the effect of irradiance ($z_{iso,tw}$) is similar across the four biomes (yellow bars in Fig. 8). Without other factors, irradiance in transparent waters would lead to migration down to 480 m in the subtropical biomes where irradiance is highest, and down to 440 m in the convective biome where irradiance is lowest.

Large-scale chlorophyll shading ($\langle\Delta z_{chl}\rangle$) reinforces the weak latitudinal contrast set by irradiance, reducing migration
depth by about 100 m on annual average in the less productive subtropical gyre, while it reduces it by approximately 140-165 m in the more productive jet region and subpolar gyre, and by up to 180 m in the convective region (Fig. 8a-c, dark green bars, see in Fig. 2d-f for annual mean chlorophyll). In the convective biome, the effect of ocean vertical transport ($\langle\Delta z_{circ}\rangle$) partially offsets that of chlorophyll shading. Migrating zooplankton are transported 100 m below their isolume on annual average due

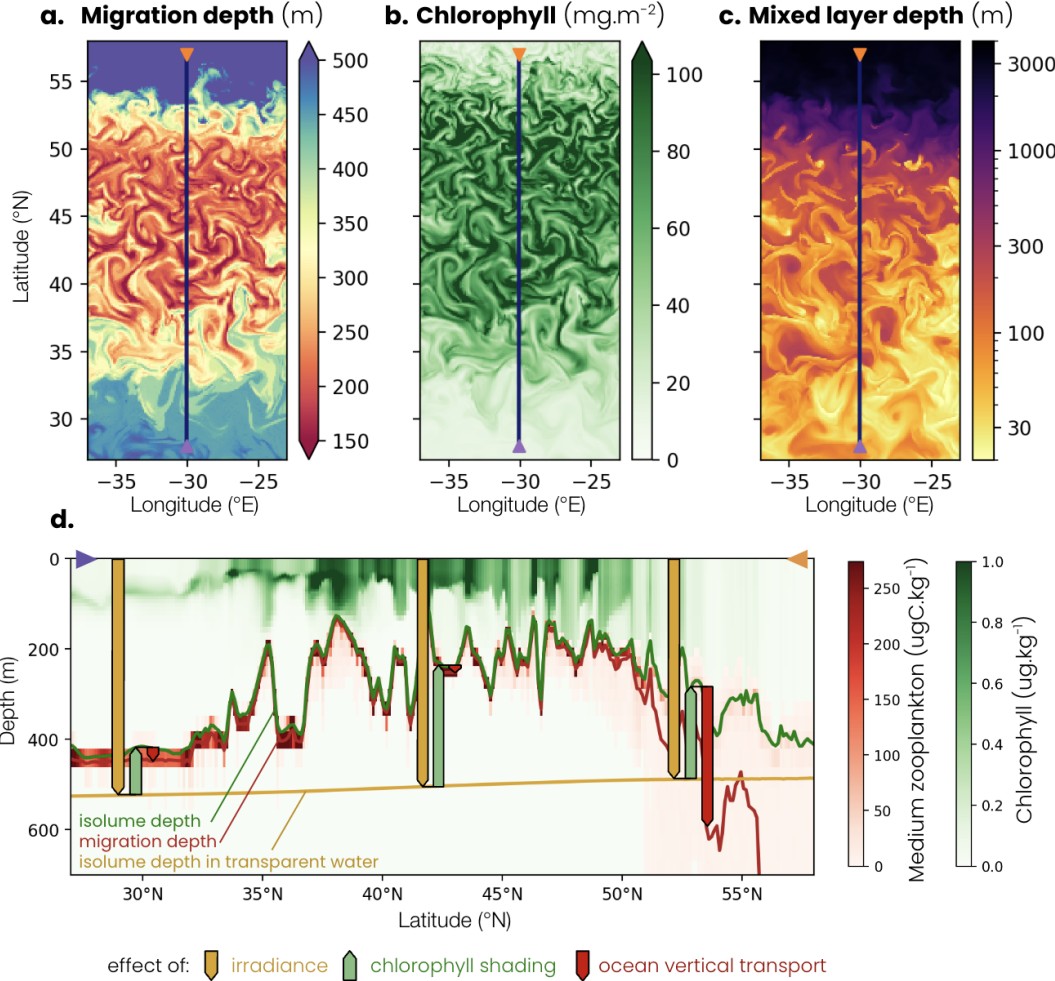

**Figure 7.** Snapshot on March 1st of (a) biomass-weighted medium zooplankton migration depth at zenith, (b) water column chrophyll content and (c) mixed layer depth. The blue line corresponds to the location of the cross-section shown on panel d. (d) Vertical cross-section of chlorophyll (green) and medium zooplankton (red) concentration on March 1st, at 30° E, between 27° N and 57° N. The depth-latitude section also shows transparent-water isolume depth (yellow line), isolume depth (green line) and migration depth (red line) at zenith. The transparent water isolume depth is a baseline controlled by irradiance intensity (yellow arrows), the difference between isolume and transparent water isolume depths is caused by chlorophyll shading (green arrows) and the difference between migration depth and isolume depth is the effect of ocean vertical transport (red arrows).

to deep convection events. In other regions, the effect of transport is not significant ($\langle \Delta z_{circ} \rangle$ lower than the model vertical

grid resolution at the depth of migration, Fig. 8a-c). The migration depth of large zooplankton is 100 m deeper than that of medium zooplankton due to irradiance, since they follow a darker isolume. However, the modulation of their migration depth




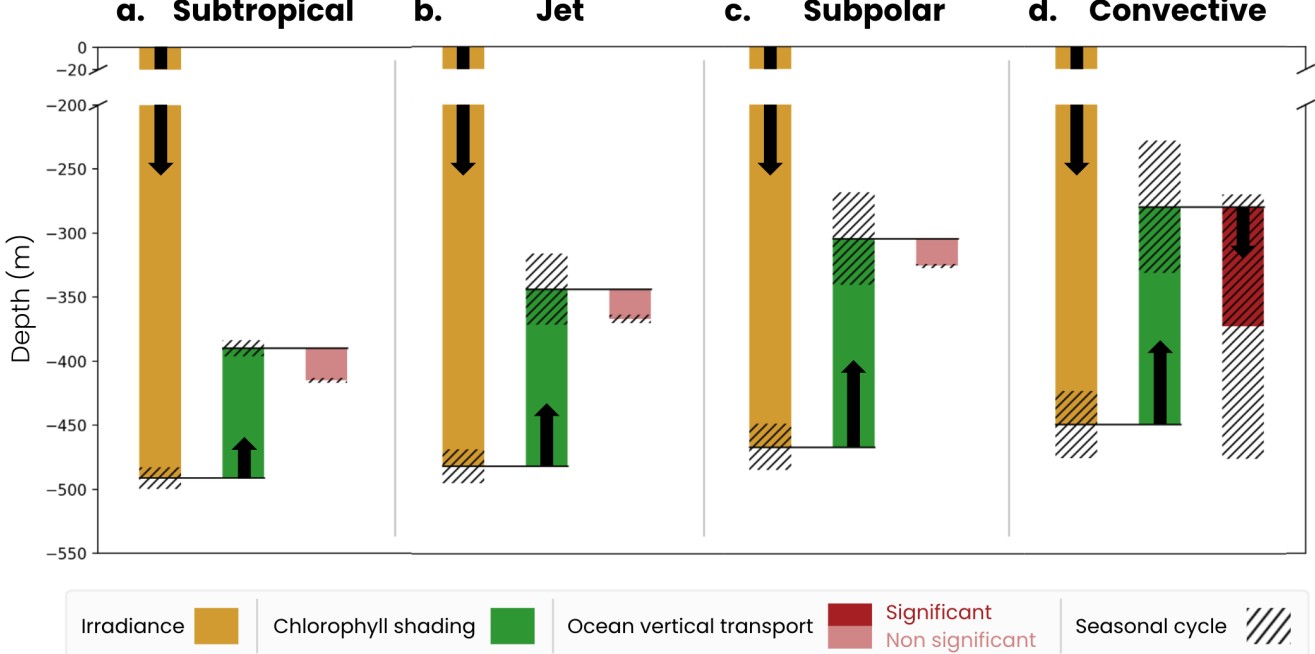

**Figure 8.** Mechanisms controlling the migration depth of medium zooplankton on annual average in the (a) subtropical, (b) jet, (c) subpolar and (d) convective biomes. The decomposition includes the effects of large-scale variations in irradiance ($z_{iso,tw}$, yellow bar), chlorophyll shading ($\langle \Delta z_{chl} \rangle$, green bar) and vertical ocean transport ($\langle \Delta z_{circ} \rangle$, red bar), and their seasonal range of variability (black hatch). The effect of transport is non-significant (light red bar) when it is smaller than the vertical resolution of the model grid at that depth.

by chlorophyll shading and transport is similar to medium zooplankton (Fig. B1). We note that the chlorophyll shading effect is not sensitive to the choice of parameters in the migration model (e.g., target isolume, migration velocity, Fig. B3).

### 3.2.3 Controls of migration depth seasonal variability

Migration depth is modulated on seasonal time-scale in response to variations in irradiance, bloom dynamics (i.e., chlorophyll absorption) and winter convection (Fig. 9). Seasonal irradiance variations deepen the migration depth in summer (deep $z_{iso,tw}$) and decrease it in winter (shallow $z_{iso,tw}$) compared to its average annual effect (Fig. 9, yellow line). The amplitude of seasonal variations in the irradiance effect increases with latitude (subtropical: 25 m, jet: 40m , subpolar: 57 m, convective: 85 m) due to greater variations in the solar incidence angle.

Large-scale chlorophyll shading considerably reduces migration depth during the spring phytoplankton bloom when chlorophyll content is highest (Fig. 9, green line). In productive regions, this reduction in the migration depth is most pronounced between February and May, and intensifies with latitude due to a more pronounced and prolonged phytoplankton bloom. The seasonal reduction in migration depth attributable to chlorophyll shading reaches 190 m in the jet region, 235 m in the subpolar gyre, and 300 m in the convective region (Fig. 9b-d). In contrast, there is no seasonal variations in the chlorophyll shading



in the subtropical gyre (Fig. 9a). Large-scale vertical transport deepens the migration depth in the convective region during convective events between January and April, down to -340 m at its peak (Fig. 9, red line). This effect is negligible at other times and in other regions.

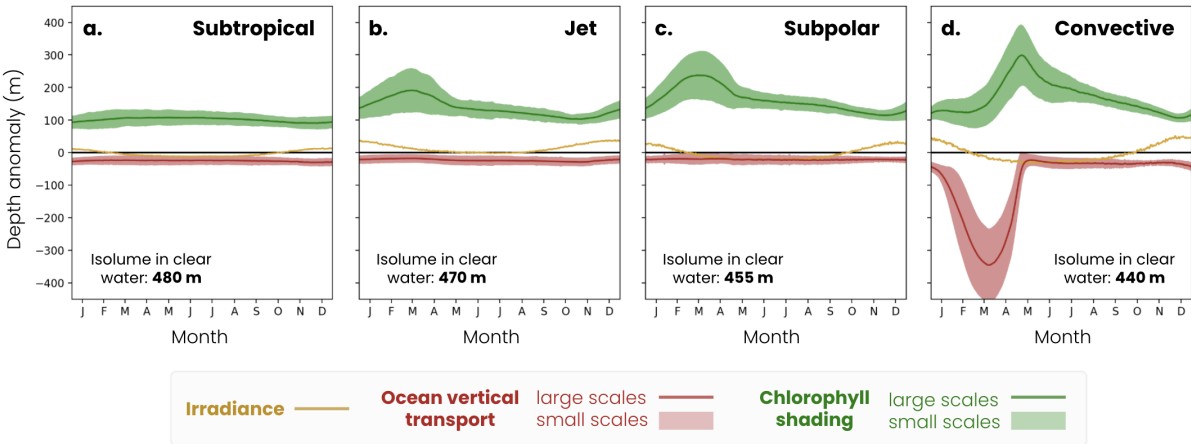

**Figure 9.** Mechanisms controlling the medium zooplankton migration depth seasonal (solid lines) and fine-scale (<100 km, shading envelope) variability in the (a) subtropical biome, (b) jet, (c) subpolar and (d) convective biomes for the irradiance ($z_{iso,tw}$ in yellow), chlorophyll shading ($\langle \Delta z_{chl} \rangle$ and $\Delta z'_{chl}$ in green) and vertical transport ($\langle \Delta z_{circ} \rangle$ and $\Delta z'_{circ}$ in red) effects. Contributions are relative to the baseline depth set by mean annual irradiance (isolume in transparent water, decreasing from 520 m in the subpolar gyre to 480 m in the convective region, see in Figure 8).

### 3.2.4 Controls of migration depth spatial variability at fine-scales

Within biomes, patchiness in phytoplankton and mixed layer depths introduces strong variations in the chlorophyll shading and transport effects ($\Delta z'_{chl}$ and $\Delta z'_{circ}$), and modulate the migration depth at fine spatial scales as shown by the snapshots from March 1st (Fig. 7a-c). The amplitude of this fine-scale variability depends on the biome and the time of year, with stronger fine-scale variability found in late winter and spring in the jet, subpolar and convective biomes (Fig. 9). On average, the chlorophyll fine-scale effects have a larger impact on migration depth than the fine-scale effects of transport. The only exception is in the convective region, where chlorophyll and transport fine-scale effects are similar.

During the spring bloom, fine-scale variations in chlorophyll and the associated shading (Fig. 9, green filling) modulate the migration depth by $\pm$ 67 m in the jet, $\pm$ 75 m in the subpolar gyre and $\pm$ 100 m in the convective region (Fig. 9b-d, green shading) and are smaller during the rest of the year (less than $\pm$ 30 m). In other words, this fine-scale effect can create migration depth contrasts of up to 200 meters between two spots 10-20 km apart during spring bloom in productive regions, reinforcing or offsetting the large-scale chlorophyll shading effect. This fine-scale chlorophyll shading effect is however much weaker ($\pm$ 20 m) and shows little seasonal variations in the subtropical gyre waters containing less chlorophyll (Fig. 9a).



In the convective region, the fine-scale effect of transport (Fig. 9, red filling) is high from January to April, and modulates migration depths up to ± 135 m, due to local changes in the mixed layer depth (Fig. 7c). During the rest of the year and in other regions, the fine-scale ocean transport effect is not significant. Thus, although these fine-scale effects are small on an annual average (less than ± 40 m), they can significantly modulate the migration depth from late winter to spring.

## 4   Conclusions and Discussions

In this study, we have developed a mechanistic zooplankton vertical migration model, which we integrated into a large-scale ocean biogeochemical model and implemented in an idealized, high-resolution double gyre physical framework. This model reproduces the large-scale contrasts in zooplankton migration depth observed between the North Atlantic subtropical and subpolar biomes (480 m vs. 380 m) (Bianchi et al., 2013b). It also captures seasonal variations similar to those observed by Hobbs et al. (2021), with a migration depth decreasing by around a hundred meters during the phytoplankton bloom. Furthermore, the model depicts variations in migration depth on fine spatial scales (< 100 km) consistent with those observed across a front by Powell and Ohman (2015), with migration depths varying by up to 200 m between points 10-20 km apart.

In the model, these large-scale, fine-scale and seasonal variations in migration depth are largely caused by the shading of chlorophyll contained in the water column, decreasing the depth of the isolume targeted by migrating zooplankton. Chlorophyll shading leads to differences in migration depth of about 60 meters on annual average between the subtropical biome (low productivity / low chlorophyll shading) and the subpolar biome (high productivity / high chlorophyll shading), and up to 150 meters during the spring bloom. In contrast, variations in migration depth caused by differences in surface irradiance between biomes are relatively small (<40 m), although we note that seasonal variations within biomes are significant (60 to 85 meters in the subpolar and convective biomes). We find that large-scale difference in chlorophyll shading effect across biomes can be reinforced or offset by 100 meters locally by fine-scale ocean dynamics associated with eddies, fronts that lead to patchy chlorophyll distribution (Gaube et al., 2014). This patchiness in chlorophyll is consistent with prior that showed fine-scale dynamics could create strong horizontal gradients in chlorophyll content through horizontal stirring (e.g., Martin, 2003), or through the stimulation of phytoplankton growth by local vertical supply of nutrients and changes in light exposure (e.g., Lévy et al., 2018). Also, contrasts in migration depth caused by differences in surface irradiance between biomes are relatively small (<40 m) but seasonal variations within biomes are significant, ranging from 60 to 85 meters in the subpolar and convective biomes.

These results suggest that the ability of migrating zooplankton to sequester carbon in the interior ocean could be limited by chlorophyll shading on both large-scale and fine-scale. Productive chlorophyll-rich regions, such as the subpolar gyre at large scale or phytoplankton-rich filaments at fine scale, provide abundant food for zooplankton and are the regions where migrating zooplankton export the most carbon (Lévy et al., 2018; Mangolte et al., 2023; Kaiser et al., 2021; Aumont et al., 2018; Archibald et al., 2019; Nowicki et al., 2022). However, it is also in these regions that chlorophyll shading reduces migration depth the most. The carbon exported by migrating zooplankton in productive regions could therefore be sequestered



for a shorter time, as the shallower the migrating zooplankton excrete or respire carbon, the faster it takes for ocean circulation to bring it to the surface, reducing the time it is potentially sequestered in the ocean interior (Boyd et al., 2019).

The model suggests that large-scale ocean transport can further modulate the migration depth, but this effect is confined to deep convective events during which zooplankton can be transported up to 330 m below their isolume. In nature, this process could potentially influence zooplankton migration in the Irminger, Greenland or Labrador Seas, where deep convection events occur (Våge et al., 2009; Piron et al., 2017; Sgubin et al., 2017). Our results also suggest that fine-scale ocean dynamics, restratifying the mixed layer and causing patchiness in ocean transport (Mahadevan, 2016; Resplandy et al., 2019), can lead to

migration depth differences up to 250 m between two locations 10-20 kilometers apart in the convective biome. However, these results are probably overestimated for two reasons. First, our idealized wind creates an intense downwelling in the convective biome at the northern boundary of the model, leading to the convection of the entire water column (MLD > 3500 m, Fig. 2a,c, Fig. A4c), while mixed layer depths observed in the Labrador, Irminger and Greenland Seas are around 200-500 m on average and up to 1500-2000 m at most (Holte et al., 2017; Gonzalez-Pola et al., 2020). Second, in nature, part of the zooplankton

community stops diel vertical migration to enter into diapause under the permanent thermocline (600-1,400 m) in winter in the North Atlantic (Jónasdóttir et al., 2015), a process not accounted for in our model. At these depths, zooplankton in diapause could be less affected by deep convection. Therefore results in the convective biome should be considered with caution.

    Existing biogeochemical models that include zooplankton vertical migration primarily use light to control diel migration patterns (Bianchi et al., 2013b; Aumont et al., 2018; Archibald et al., 2019; Nowicki et al., 2022). In contrast, our model in-

troduces a novel prey-driven redistribution mechanism at night, where zooplankton adjust their vertical positions to optimize feeding opportunities by aligning with depths of higher prey concentration, which more accurately captures the critical role of prey availability in shaping zooplankton behavior (Dini and Carpenter, 1992; Beklioglu et al., 2008). However, the model is limited by the lack of predator-led controls, partly because key zooplankton predators are not explicitly represented. This omission could oversimplify zooplankton migration behavior, especially in regions where predator pressure is significant (Bol-

lens and Frost, 1989). In reality, zooplankton DVM is influenced by a balance between avoiding predators and optimizing feeding, with predator pressure often modulating the depth and timing of migrations (Hays, 2003; Bandara et al., 2021). Future improvements could aim to integrate predator-prey interactions more comprehensively to enhance the realism and applicability of the model across various marine environments (Pinti et al., 2019).

    This study highlights the role of chlorophyll in setting the zooplankton migration depth in the North Atlantic. Other studies

have underlined the importance of other environmental variables such as moonlight (Last et al., 2016) or ice cover (Petrusevich et al., 2016; Flores et al., 2023) modulating the light landscape and the isolume depth in the Arctic regions, cloud shadow in the subpolar North Pacific (Omand et al., 2021), or the oxygen concentration limiting zooplankton migration in the upper margin of oxygen minimum zones (Bianchi et al., 2013a). Although estimates are uncertain, climate models predict that most of these parameters will evolve in response to climate change (Kwiatkowski et al., 2020; Notz and Community, 2020; Vignesh et al.,

2020).A decrease in ocean productivity or ice cover could deepen the isolume and ocean deoxygenation could alter migration depth in oxygen-limiting regions. Moreover, zooplankton size is one of the main features controlling their migration depth (Ohman and Romagnan, 2016). Warmer and less productive waters tend to support smaller zooplankton communities (Barton



et al., 2013; Brun et al., 2016) which could dive to shallower depths. The diversity and complexity of these processes make
it difficult to predict the evolution of zooplankton migration depths in the future around the globe. However, COBALTv2-
DVM captures all these mechanisms, and its implementation in a global ocean model could provide a way of quantifying their
importance.

The patchiness of zooplankton migration depth caused by chlorophyll in spring makes it difficult both to observe migration
patterns and their consequences in the ocean. The patchiness of phytoplankton, which has been discussed for a long time
(Mackas et al., 1985; Abraham, 1998; Martin et al., 2002), is easily observable from satellites (Lehahn et al., 2018; McClain
et al., 2022). However, chlorophyll heterogeneity spreads to the migration depth of zooplankton, which is difficult to measure.
Currently, migration depth is mainly derived from ADCP profiles (Bianchi et al., 2013b; Riquelme-Bugueño et al., 2020;
Cisewski et al., 2021) or net measurements (Steinberg et al., 2008; Conroy et al., 2020). These observations are expensive
because they are carried out from ships during oceanographic campaigns. Consequently, they are too sparse to resolve and
isolate signals of large- and fine-scale spatial variability, as well as the seasonal cycle of migration depth in many regions of
the ocean (Bianchi et al., 2013a). Yet, migration signatures have already been observed by ARGO floats (Fig. S3 in Boyd et al.,
2019). Developing algorithms to detect these migrations in ARGO floats, as has already been done to detect subduction events
(Llort et al., 2018) or deep chlorophyll maxima (Cornec et al., 2021), could increase the number of observations available.
More observations of zooplankton migration are crucial to better understand its variability, constrain it in ocean models and
assess its impact on biogeochemical cycles.

*Code availability.* The COBALTv2-DVM model and idealized double gyre configuration presented in this paper are available on github at
https://github.com/mpoupon/MOM6_Double_Gyre.

*Author contributions.* Conceptualization: MP, LR, JL. Data curation: MP. Formal analysis: MP. Funding acquisition: LR. Investigation: MP.
Methodology: MP, LR. Software: MP, JL, JG, NZ. Supervision: LR, JL, CS. Visualization: MP. Writing – original draft: MP, LR. Writing –
review and editing: MP, LR, JL, JG.

*Competing interests.* The authors declare that they have no conflict of interest.

*Acknowledgements.* We thank John Dunne for conducting the GFDL internal review. The study has been supported by the National Science
Foundation (Eddy effects on Biological Pump award number 2023108).



## Appendix A: Materials and Methods

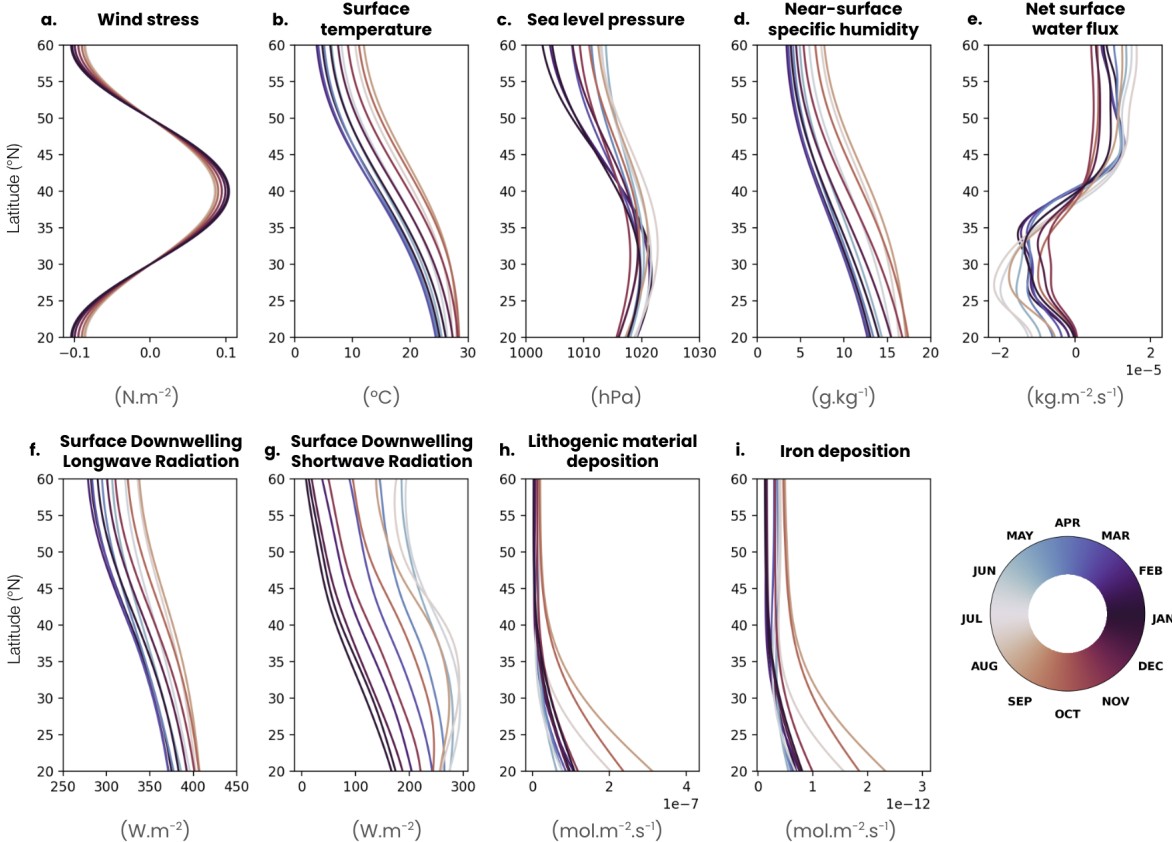

**Figure A1.** Surface forcing seasonal cycle in the double gyre model. (a) Zonal wind stress following an idealized sinusoidal profile varying by 10% in amplitude according to a seasonal cycle. (b) Atmosphere surface temperature, (b) Sea level pressure, (d) Near-surface specific humidity and (e) precipitation derived from the JRA55-do atmospheric reanalysis (Tsujino et al. 2018), with a 3-h time resolution averaged over the period 1958-2020, and zonally of the North Atlantic Ocean (longitude = -65° E to -20° E, latitude = 20° N to 60° N). Surface downwelling (f) longwage and (g) shortwave radiation calculated as the zonal mean of the the ERA5 atmospheric reanalysis (Hersbach et al. 2020) averaged over the 1979-2022 period and zonally over a band located in the middle of the North Atlantic (longitude = -35° E to -33° E, latitude = 20° N to 60° N). Deposition of (h) lithogenic material and (i) iron deposition are calculated from the last 30 years of a GFDL-ESM4 pre-industrial control simulation spanning from 1850 to 2014 in the same way as the physical forcings (Dunne et al. 2020). Each profile is to the monthly average of the variables and the color corresponds to the month.



**Figure A2.** Comparison between the double gyre model and the World Ocean Atlas 2018 (WOA18). Zonal mean of (a,b) nitrate concentration, (c,d) phosphate concentration, (e,f) silica concentration, (g,h) oxygen concentration, (i,j) salinity and (k,l) temperature in the double gyre model (top row) and WOA18 (bottom row, -60° E to -20° E and 20° N to 60° N).





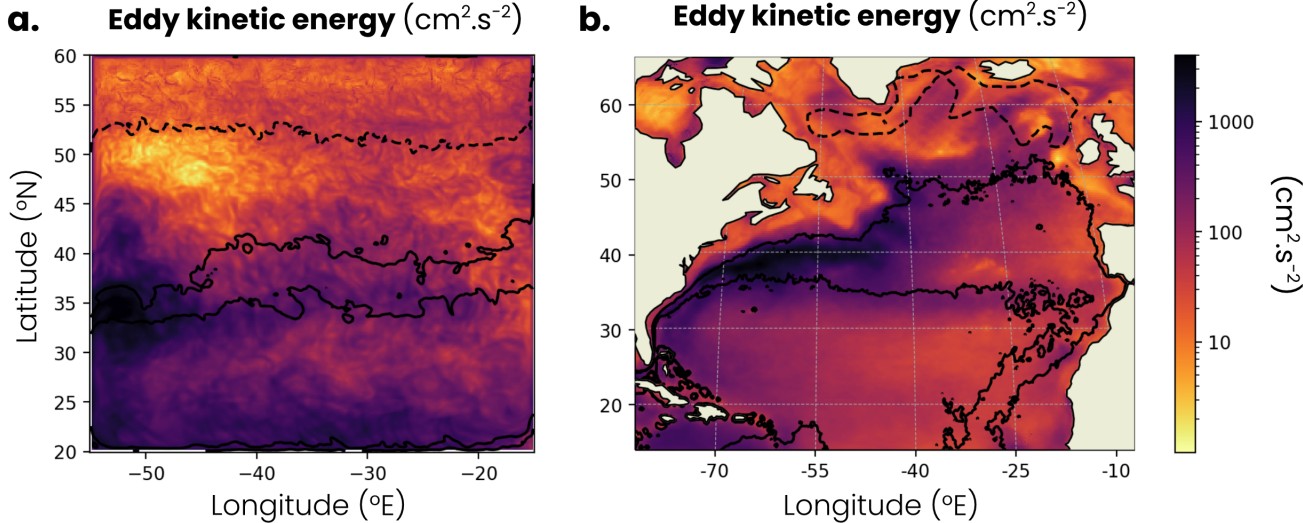

**Figure A3.** Map of annual mean surface eddy kinetic energy (a) in the double gyre model and (b) derived from sea surface height observations between 1993 and 2021 (Archiving, Validation and Interpretation of Oceanographic Satellite Data - AVISO). The subtropical, jet and subpolar biomes are delimited by solid black lines (mean surface chlorophyll concentration = 0.15 and 0.35 $mg.m^{-3}$), while the subpolar and convective biomes are delimited by a dashed line (mean MLD = 200 m).



**Figure A4.** Seasonal cycle of (a,b) surface chlorophyll concentration and (c,d) mixed layer depth in (a,c) the double gyre model and (b,d) observations as a function of latitude. The seasonal cycle is calculated as the zonal mean over the whole domain in the model and in a North Atlantic box (longitude = -65° E to -20° E, latitude = 20° N to 60° N) in the observations. Observations of mixed layer depths are derived from a compilation of hydrographic sections between 1941 and 2002 (de Boyer Montegut et al. 2004), surface chlorophyll from satellite observations between 1997 and 2020 (Sathyendranath et al. 2019).





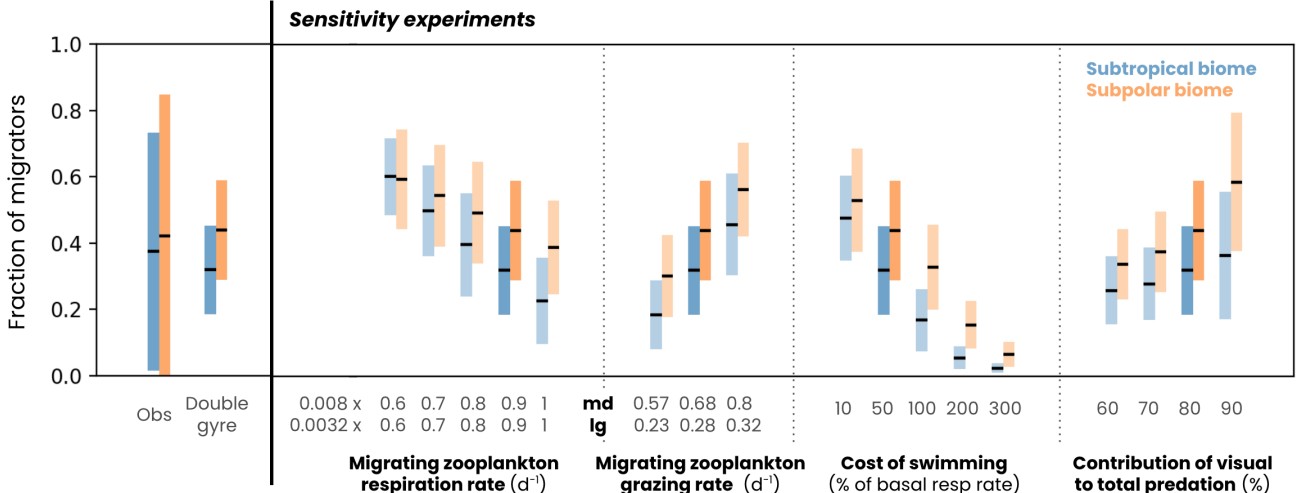

**Figure A5.** Fraction of zooplankton migrating in the subtropical (blue) and subpolar (orange) biomes in observations (from the COPEPOD database and compiled by Aumont et al. 2018), the double gyre model and a series of sensitivity experiments. The black line corresponds to the mean and the bars to two standard deviations around the mean. The dark blue and orange bars correspond to the simulation described in the paper and the light blue and orange bars to the parameter sensitivity experiments.





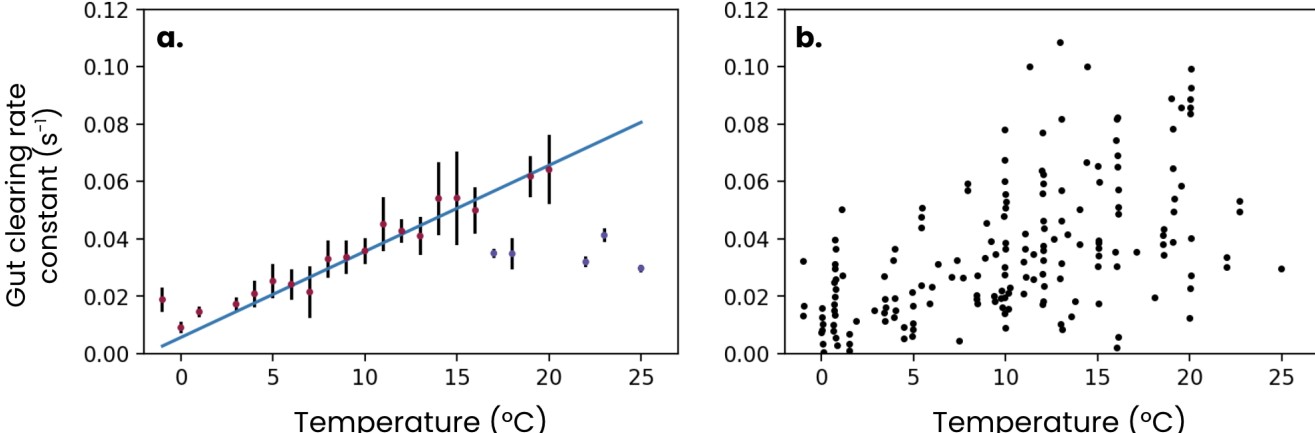

**Figure A6.** Observations of zooplankton gut clearance rate constant versus temperature, (a) binned for each degree of temperature or (b) raw. Digitised data extracted from Irigoien (1998).



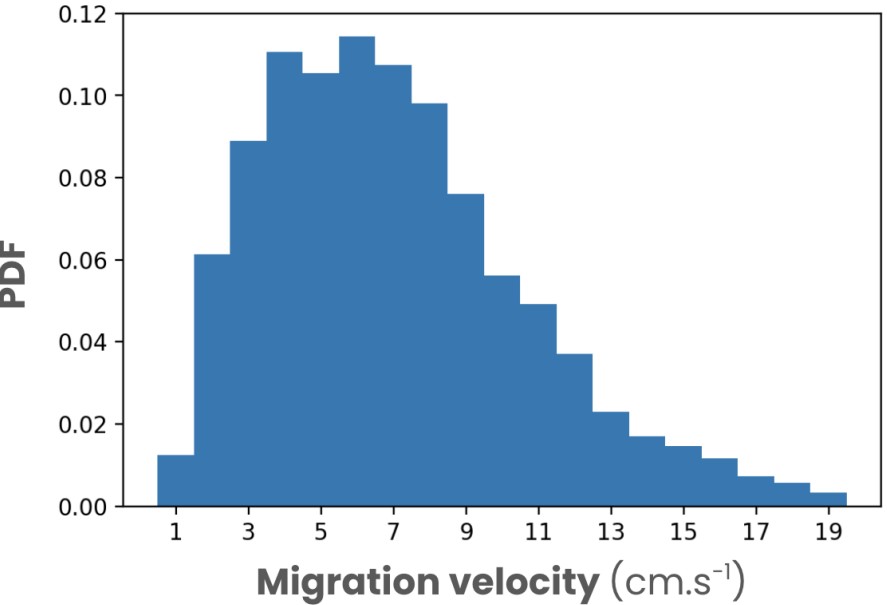

**Figure A7.** Global distribution of vertical migration velocities of zooplankton measured from global acoustic data by Bianchi and Mislan (2016).



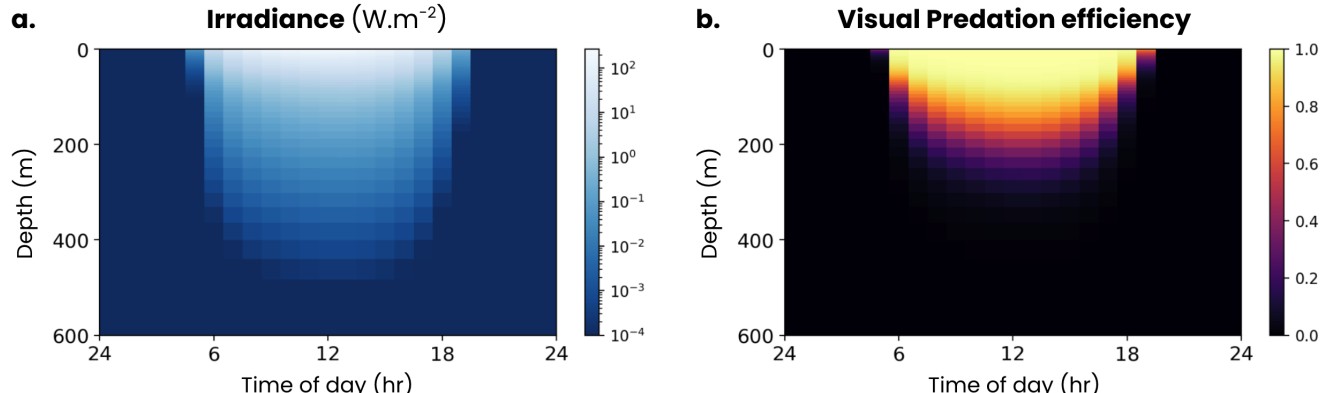

**Figure A8.** Mean daily cycle of the (a) irradiance and (b) visual predation efficiency vertical distribution in the double gyre model.





**Table A1.** Major parameters and associated values used in the physical ocean (MOM6) component of the model

| Parameter | Spin-Up | Simulation | References |
|---|---|---|---|
| Horizontal grid and resolution | 42 x 42, 85 km | 362 x 362, 9.4 km | |
| Vertical coordinate | 45 layer hybrid $z^*$-isopycnal | 75 layer hybrid $z^*$-isopycnal | (Adcroft et al. 2019) |
| Number of CPUs | 60 | 1024 | |
| Baroclinic and Biogeochemical time steps | 1800 s, 1800 s | 300 s, 600 s | |
| Planetary boundary layer parameterization | ePBL | ePBL | (Reichl and Hallberg. 2018) |
| Subgrid Mesoscale EKE parameterization | MEKE | No | (Hallberg et al. 2013) |
| Submesoscale restratification parameterization | No | Frontal length = 1500 m | (Fox-Kemper et al. 2008) |
| Background kinematic viscosity | $K_V = 10^{-4}$ m$^2$.s$^{-1}$ | $K_V = 10^{-6}$ m$^2$.s$^{-1}$ | |
| Background diapycnal diffusivity | $K_D = 10^{-4}$ m$^2$.s$^{-1}$ | $K_D = 10^{-6}$ m$^2$.s$^{-1}$ | |
| Horizontal viscosity | Laplacian ($A_H = 10^5$ m$^2$.s$^{-1}$) $A_H = 10^5$ m$^2$.s$^{-1}$ | Smagorinsky biharmonic Smagorinsky coefficient = 0.015 Resolution-dependent = 0.01 $\Delta_x^3$ m$^4$ s$^{-1}$ | (Griffies et al. 2000) |
| Opacity Scheme | 3-band with chlorophyll | 3-band with chlorophyll | (Manizza et al. 2005) |





**Appendix B: Results**

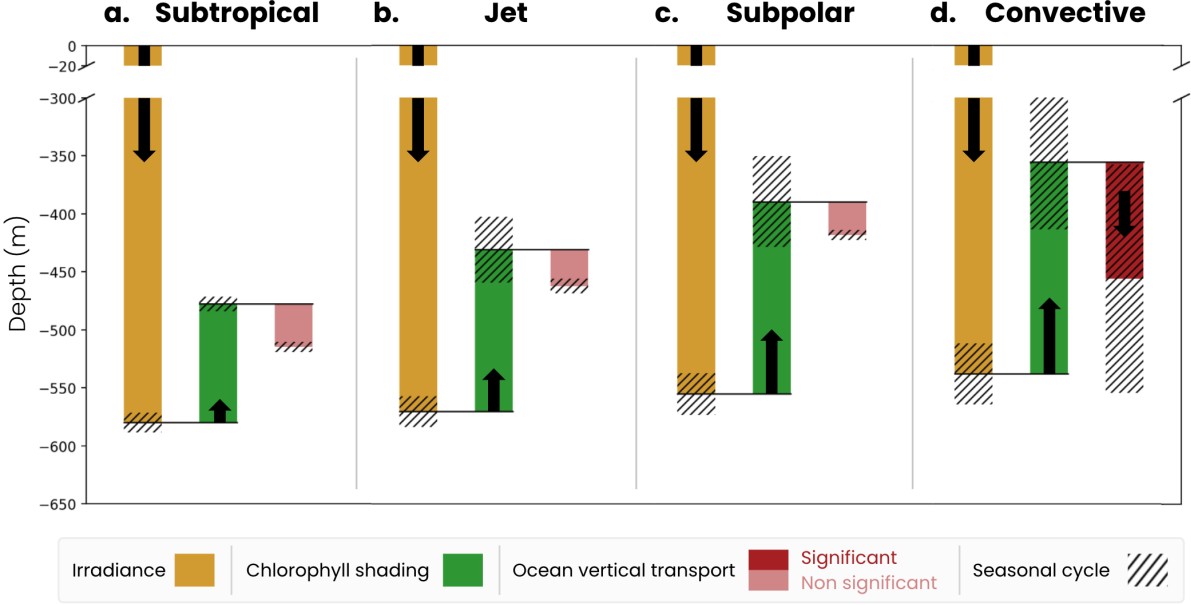

**Figure B1.** Decomposition of the mechanisms controlling the migration depth of large zooplankton in the (a) subtropical, (b) jet, (c) subpolar and (d) convective biomes. The decomposition includes the effects of large-scale variations in irradiance (yellow bar), chlorophyll shading (green bar) and vertical ocean transport (red bar), and their seasonal range of variability (black hatch). The effect of transport is non-significant (light red bar) when it is smaller than the vertical resolution of the model grid.



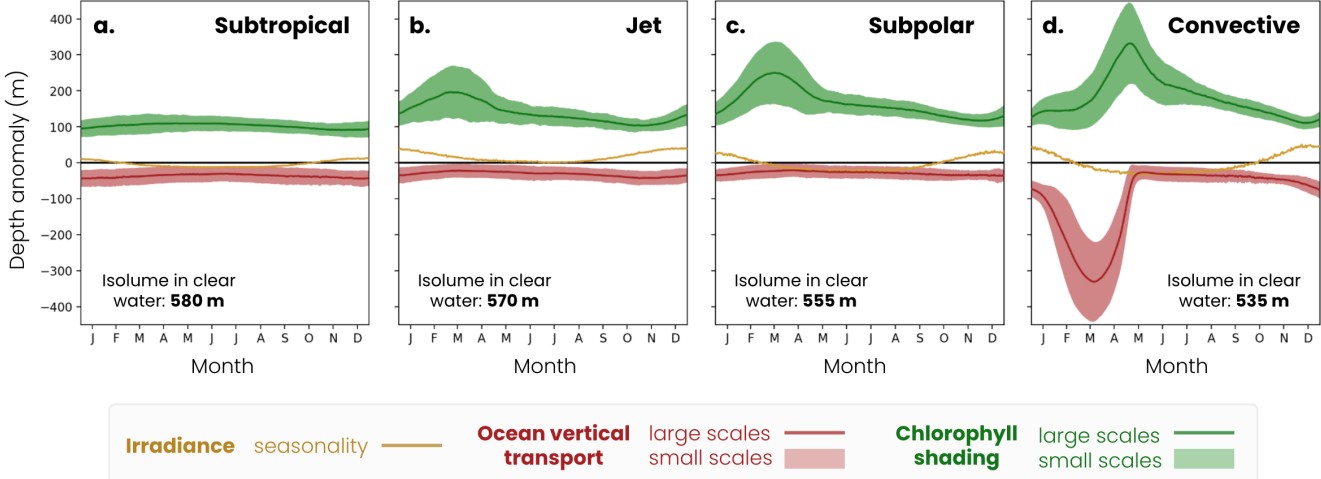

**Figure B2.** Seasonal decomposition of the mechanisms controlling the medium zooplankton migration depth in (a) the subtropical biome, (b) the jet, (c) the subpolar biome and (d) the convective biome. The lines are the effects of seasonal variations in irradiance (yellow line), large-scale chlorophyll shading (green line) and large-scale vertical transport (red line) relative to a baseline depth set by mean annual irradiance (isolume in clear water, decreasing from 520m in the subpolar gyre to 480m in the convective region). Large-scale effects are superimposed by small-scale spatial variability in chlorophyll shading (green fill) and vertical transport (red fill) induced by eddies and fronts.





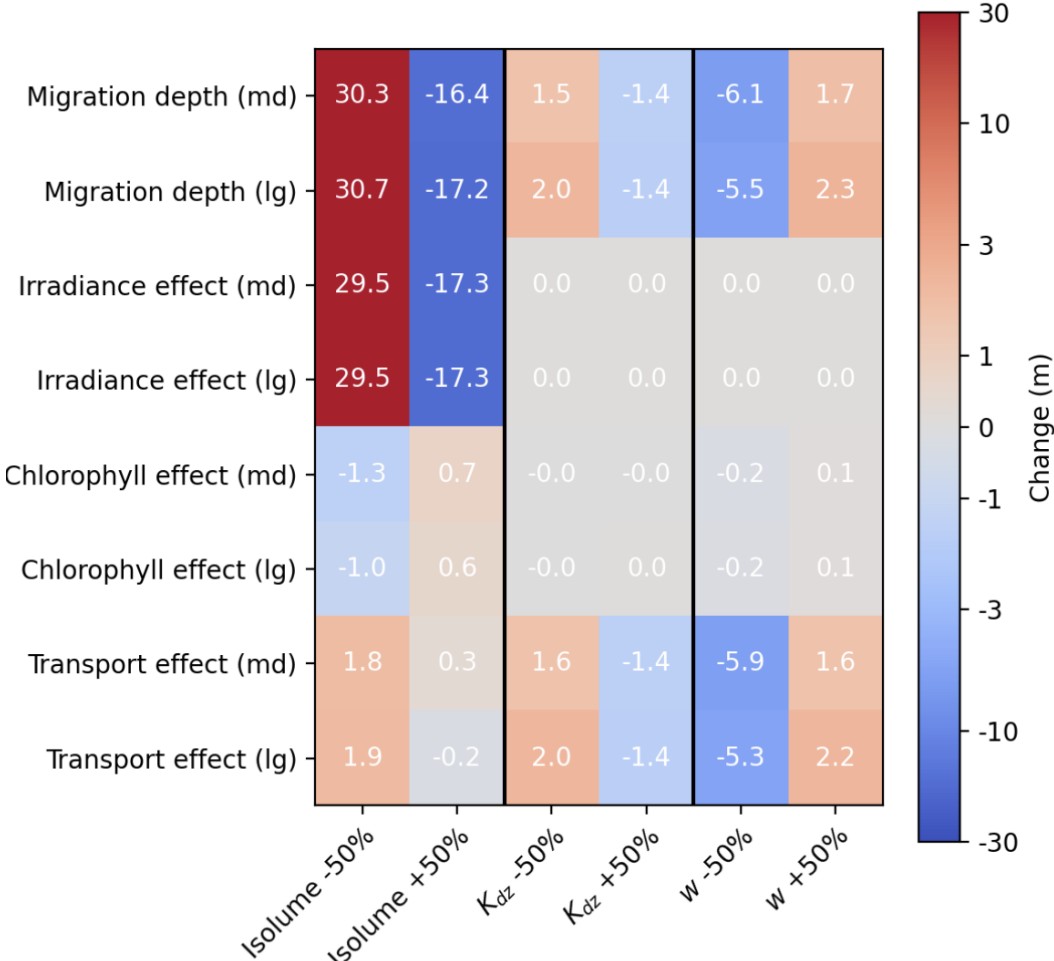

**Figure B3.** Sensitivity experiment on zooplankton migration depth. Change in migration depth, effect of irradiance, chlorophyll and transport for medium (md) and large (lg) zooplankton in response to a 50% decrease or increase in target isolume, half-saturation distance ($K_{dz}$) and migration velocity ($w$). A positive anomaly indicates a deeper migration depth and a negative anomaly a shallower migration depth.



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
