# Peer review of "Chlorophyll shading reduces zooplankton diel migration depth in a high-resolution physical biogeochemical model."

_EGUsphere, 2024_

## Author Comment (AC1)

We thank the reviewers for their constructive comments and suggestions. We have addressed all the points raised and implemented the suggested changes, including clarifying the model description, moving the model evaluation to the result section and correcting a few typos. We hope that the manuscript is now ready for publication in Ocean Science.

Our *detailed* response *to the* reviewers' questions and comments can be found below. All page/line/reference/figure numbers refer to the tracked manuscript. Reviewers' comments are in regular text, our responses are in blue and *new text from the manuscript is in blue italics.*

**Review 1**

**Specific comments:**

A key review study in the field that I think deserves mention is Nandara et al 2021, as it reviews sampling, observation and tracking simulation methods, while also emphasizing the importance of integrative approaches.

Kanchana, et al. "Two hundred years of zooplankton vertical migration research." *Biological Reviews* 96.4 (2021): 1547-1589.

**Answer:** Indeed this work by Bandara et al. (2021) is relevant. We cite this article in the context of mechanisms controlling zooplankton migration on lines 40 and 481. We have added a mention of this article in the discussion to emphasize integrative approaches (line 507).

The model considers two zooplankton sizes, both of which fall within the mesozooplankton size fraction, with copepods being key representatives. However, could the model be applied to or used with microzooplankton ? I imagine the complexity of including both groups lies in the differences in egestion and assimilation of various elements, as well as other factors.

**Answer:** Yes, the model could technically be used to represent the vertical migration of microzooplankton, after adjusting the parameters controlling their physiology and migration patterns. For example, in our model, we already represent two sizes of mesozooplankton, which differ in their light preferences, to reflect the fact that the smaller the zooplankton, the shallower their depth and the smaller their migration amplitude (Ohman and Romagnan, 2016). Similar adjustments could be made to represent the notable differences between mesozooplankton already included in the model and newly added microzooplankton. However, it is important to note that microzooplankton exhibit markedly different migration dynamics compared to mesozooplankton. When contrasting copepod migration depths (Ohman and Romagnan 2016) with that of ciliates (e.g., Wang et al. 2023), microzooplankton vertical

movement is often more subtle and constrained in scale compared to that of mesozooplankton  partly because they do not have the same swimming ability as mesozooplankton.

Ohman, Mark D., and Jean-Baptiste Romagnan. "Nonlinear effects of body size and optical attenuation on diel vertical migration by zooplankton." Limnology and Oceanography 61.2 (2016): 765-770.

Wang, Chaofeng, et al. "Diel variations in planktonic ciliate community structure in the northern South China Sea and tropical Western Pacific." Scientific Reports 13.1 (2023): 3843.

N:P=1:16 did you try varying this across different biomes?

**Answer:** The COBALTv2 model, on which COBALTv2-DVM is based, incorporates a static N:P ratio of 16:1 for mesozooplankton (note, however, that for phytoplankton and microzooplankton, the N:P is still static but differs from 16:1). This code cannot yet model a variable N:P ratio for zooplankton, however a dynamic stoichiometry has recently been implemented for phytoplankton (Hagstrom et al. 2024), which could be the basis for future developments applied to zooplankton. We would expect to observe higher N:P ratios in oligotrophic zones such as the subtropical biome than in eutrophic zones such as the subpolar biome, which could modulate the regional nutrient cycling.

Hagstrom, George I., et al. "Impact of dynamic phytoplankton stoichiometry on global scale patterns of nutrient limitation, nitrogen fixation, and carbon export." Global Biogeochemical Cycles 38.5 (2024): e2023GB007991.

I am curious about the difference in the dates for the datasets: migrating zooplankton data were collected between 2007 and 2019, while MODIS data span from 2002 to 2023. Given that 2023 was a particularly warm year, did including or excluding this year make a difference? Or was it clearly not affecting the seasonal variations?

**Answer:** We have tested the sensitivity to this particularly warm year as suggested by the reviewer. As shown in the figure below, the productivity estimates from MODIS measurements for 2023 are slightly lower than seasonal climatology, however the differences between climatologies calculated with and without 2023 are very small (subtropical: -0.55%, subpolar: -0.37%) and does not clearly affect seasonal variations.

[Figure]

Line 161: Nitrate and phosphate values?

**Answer:** We have clarified the values as follows:

*L301-303: "In both the observations and model, the subtropical anticyclonic gyre (elevated sea surface height, Fig. 1a) is oligotrophic at the surface, with nitrogen concentrations averaging 0.29 and 0.28 umolN kg⁻¹ respectively, and phosphorus concentrations averaging 0.03 and 0.06 umolP kg⁻¹ in the first 100 meters."*

There is some inconsistent formatting in the reference section.

**Answer:** We have not found these inconsistencies but will be happy to modify the reference section as needed.

---

## Author Comment (AC2)

We thank the reviewers for their constructive comments and suggestions. We have addressed all the points raised and implemented the suggested changes, including clarifying the model description, moving the model evaluation to the result section and correcting a few typos. *We hope that the manuscript is now ready for publication in Ocean Science.*

Our *detailed* response *to the* reviewers' questions and comments can be found below. All page/line/reference/figure numbers refer to the tracked manuscript. Reviewers' comments are in regular text, our responses are in blue and *new text from the manuscript is in blue italics.*

**Review 2**

My main suggestion is to move Section 2.3 from the Methods to the Results. This Section is basically detailing results from the model and validating it against a variety of observations, which does not properly belong to the Methods. It could easily form the beginning of Section 3, Results, and lead into the current Section 3.1. This would lead to a smoother flow for the reader.

**Answer:** We thank the reviewer for this suggestion. To smooth the flow we have moved section 2.3 (formerly named: Model biome definition and biogeochemical evaluation, now: Contrasted biogeochemical seasonal cycles) to the results section. We have merged it with the old section 3.1 (North Atlantic migrating zooplankton) to form the next section:

- 3.1 - North Atlantic biophysical dynamics
    - 3.1.1 - Contrasted biogeochemical biomes
    - 3.1.2 - North Atlantic migrating zooplankton

Line 101: Model configuration. Since MOM6 is based on a generalized vertical coordinate, it may be worth clarifying how the vertical coordinate is handled in this configuration.

**Answer:** The vertical coordinates of the model are Z*, i.e. the vertical levels adjust dynamically to the free surface height changes. We added this information to the model description section:

*"The domain is a square of 3570 km side length, 4000 m depth, 9 km horizontal resolution, 2-5 m vertical resolution between 0 and 100 m depth, 10-100 m between 100 and 1000 m depth and up to 250m below. **The model uses Z\* vertical coordinates, with depths dynamically adjusting to accommodate changes in the free surface height (Adcroft and Campin, 2004).**"*

Forcings (Section 2.1.2). The forcings used to set up the two-gyre configuration seem appropriate. It may be helpful to clarify whether the light forcing is temporally synchronous

across the whole basin on daily timescales, or if the timing of daily cycle changes differs going east to west (as in the real world).

Similarly, hourly light forcing is probably accurate for the first-order questions investigated by the Authors, although it may miss or misrepresent "crepuscular" periods, i.e., the periods of light before sunrise and after sunset, which generally last less than 1 hour, during which there is enough light to influence the behavior of migrating organisms. This could be noted.

**Answer:** We thank the reviewer for these suggestions. Light forcing is temporally synchronous across the whole basin on daily time scales. We acknowledge that the model is more appropriate for dealing with first-order questions rather than crepuscular situations requiring finer light resolution, although the model linearly interpolates the light values for each time step (every 5 min) between the forcing values available every hour. These suggestions have been integrated into the forcing description section:

*"This radiative forcing, like all the other surface forcings, is temporally synchronous across the whole basin on daily time scales and linearly interpolated every 5 minutes, between available forcing values. This resolution enables our model to accurately represent the mean daily light cycle, although it may not perfectly resolve the crepuscular period, when zooplankton begin to migrate."*

Section 2.2, Observations. It may be valuable to note that observations of DVM biomass are particularly difficult and observational results are likely method- and community- dependent, with different approaches observing different migrating organisms / communities. E.g., acoustics, from which many metrics of DVM behavior are obtained, are biased towards strongly sound-scattering organisms (e.g., gas-bladdered fish); net samples target specific sizes, with under-sampling of organisms smaller than the net mesh, and of large size classes because of net avoidance. The lidar-based approach by Behrenfeld et al., 2019 is very promising, and makes for an excellent addition to Fig. 2, but it is also highly uncertain and potentially biased by issues in both the lidar and net sample data that are used to generate it. That said, the broad agreement of model, lidar-based, and net-based observations (at BATS), across meridional gradients, as shown in Fig. 2, is very encouraging.

**Answer:** We have completed the observation description section to highlight the challenge of getting these observations and their potential biases:

*"Note that these observations are particularly rare and challenging to collect, and may be subject to measurement method bias. Acoustic methods can be biased towards highly sound-scattering animals (Flagg and Smith, 1989), net measurements capture neither animals smaller than the mesh, nor larger ones able to avoid the net,*

*while LIDAR data can be biased towards larger plankton and miss biomass capture of smaller ones."*

Section 2.4.1, Migration Model. I appreciate the detail of the parameterization description, and the improvements to previous formulations, such as the smooth swimming speed function, and the redistribution of migrating organisms following the prey at night. I was curious to hear whether you tried to re-distribute organisms based on non-migrating zooplankton, though I imagine that would provide nearly identical results.

**Answer:** We considered redistributing migrating zooplankton following the vertical distribution of non-migrating zooplankton, but did not try it for several reasons. Such a method would have required the choice of arbitrary criteria for which we had few constraints, such as the time at which to perform the redistribution. A redistribution would have lost the mechanistic and realistic representation of migration by creating a discontinuity in the migration trajectory, which could have had repercussions on biogeochemical fluxes. This solution would also have been computationally more costly than adding a simple vertical migration velocity to zooplankton advection.

I also suggest adding some detail to this section, as I was slightly confused by how the transition between a vertical velocity that targets the optimal isolume during the day and the redistribution based on the excess zooplankton at night is handled. I imagine that at night, when the isolume is absent, the only velocity applied is that based on zooplankton excess, which would drive upward migrations until migrating zooplankton match the prey distribution. If there is a way to amend or expand equation (1) to include all cases, including the nighttime case, that could help clarify this point.

**Answer:** We thank the reviewer for this comment that helped clarify the model description. Indeed, this is exactly what happens. During the day, when the isolume is present, the zooplankton above it swim downwards and those below it swim upwards "to reach their prey". However, as soon as they reach the isolume, they stay at its depth, because all zooplankton above the isolume swim downwards. At night, when the isolume is absent, zooplankton swim upwards until they redistribute around their prey. The migration model description section has been modified to clarify this point by adding a term, $\eta_{iso}(z)$, to equation 1 to better describe the transition between the control of migration by light and food:

$$w(z) = \eta_{iso}(z)w_0 \frac{|\frac{1}{K_{Blue}}\ln(\frac{I_{iso}}{I(z)})|}{K_{dz} + |\frac{1}{K_{Blue}}\ln(\frac{I_{iso}}{I(z)})|}$$

The paragraph has been restructured and the following parts added:

*"$\eta_{iso}(z)$ is 1 or -1 depending on the zooplankton position (z) and controls the migration direction. When zooplankton are above the isolume depth ($z<z_{iso}$), $\eta_{iso}(z)$ = 1 and zooplankton swim downwards. When the zooplankton is below the isolume, e.g. because the isolume rises at dusk or is absent at night,* migrating zooplankton return to the surface to feed. [...]*

*To reproduce foraging behavior, migrating zooplankton distributions are redistributed according to the distribution of its preys and $\eta_{iso}(z)$ can be 1 or -1, depending on:*

$$ZooExcess(z) = \int\limits_{0}^{z} \left( \frac{Z(z')}{\overline{Z}} - \frac{P(z')}{\overline{P}} \right) dz'$$

Where Z(z') and P(z') are the concentrations of zooplankton and its prey at depth z'. Z and P is their mean concentration over the entire water column. At a given depth $z_0$, if ZooExcess(z)>0, the zooplankton is in relative excess with respect to its prey between the surface and this depth. Therefore, zooplankton dive (*$\eta_{iso}(z)$ = 1*) to restore the balance. If ZooExcess(z)<0, the zooplankton is in relative deficit to its prey. Therefore, zooplankton rise to fill this imbalance *($\eta_{iso}(z)$ = -1). Note that if the zooplankton below the isolume rises and reaches it, it remains at the isolume depth, since its migration velocity becomes zero (w(z=$z_{iso}$)=0)."*

The redistribution method is clever, but I also wonder how biologically consistent it is, since in the real world organisms respond to a combination of proximate cues and intrinsic behaviors, without a complete knowledge of the prey distribution that only a model can provide.

**Answer:** We agree that the biological reality is more complex, since migration is actually triggered by many of the cues mentioned in the introduction, some of which are not represented by ocean biogeochemical models (e.g. kairomones from predators). However, integrating this process into such complex models requires a parsimonious and computationally inexpensive formulation that necessitates a certain degree of compromise with biological reality.

To my recollection, Bianchi et al., 2012 included a biological diffusion term, mimicking random vertical movement, which prevented organisms from accumulating into a single model layer, spreading them more smoothly in the water column. It is not clear if a similar term is needed, and if it is included here. Fig. 3 shows that the migrating layers are vertically spread in a fairly realistic way. I wonder if this is an effect of numerical diffusion for vertical movement (is an implicit method used?), and if this could lead to resolution-dependent results, e.g., if time step or vertical layering were to change significantly.

**Answer:** In our model, no additional vertical diffusion has been added to reproduce random vertical movements. However, migrating zooplankton is subject to the same vertical diffusion as the other tracers in the model, which can be high in the mixed layer. In addition, an implicit method is used to solve the advection-diffusion equation, making the solution more diffusive. This limits to some extent the accumulation of zooplankton in a single layer. We did not observe any significant sensitivity of the solution to vertical resolution, but we did note variations in the solution as the temporal resolution increased, with less realistic results as the time step approached 1h (irregular migration with jumps observed). A time step shorter than 30 min is sufficient to represent regular migration without jumps.

Section 2.4.3. Likewise, the development of a physiological zooplankton model with internal pools is impressive, and may prove essential for simulating effects of DVM on material transport in the water column (i.e., active transport). The formulation appears to be sound, although a few aspects could be clarified. For example, it may be useful to clarify equation 7 to show that it is applied to determine growth as a function of kresp*Nmetab - respiration, and that the resulting "growth" term can be either a growth (positive) or a loss (negative) term. Furthermore, it may be worth stating more clearly that this complex physiological parameterization is only applied to migrating zooplankton, and that, in the absence of migrations, it should produce similar results as for non-migrating zooplankton (i.e., growth and mortality rates would be comparable).

**Answer:** We thank the reviewer for these suggestions. Indeed, "negative" growth should be interpreted as mortality. We have modified the physiological model description section to include the suggested clarifications:

*"Biomass production by anabolism (growth) is equal to the difference between the metabolic (metabolism) and catabolic fluxes (respiration). If this difference is positive, the biomass increases. If it is negative, part of the zooplankton does not have the metabolic resources to ensure its catabolic needs. This part of the zooplankton dies and its biomass is transformed into sinking detritus"*

*"Note that this complex physiological parameterization is only applied to migrating zooplankton, in order to separate fluxes temporally and therefore spatially. Without migration, this parameterization would give similar results to non-migrating zooplankton, on a daily average."*

Section 2.5.1: this is a really clever way of decomposing the effects of different DVM drivers. One question I had is whether the circulation component is mostly caused by vertical mixing (during deep convection), or if it includes horizontal transport components.

**Answer:** The circulation component is caused by vertical mixing, mainly during deep convection events in winter. Horizontal transport is too weak at migration depths to influence zooplankton position on the short time scales considered.

The match between observed and modeled DVM depth in Figure 6 is impressive. I'm not surprised by the large model variability in the sub polar sector, which may be underestimated by sparse observations in such a variable region.

**Answer:** We share the reviewer's intuition, which ties in with our last paragraph of discussions on the need to develop new methods for measuring migration depth in order to better capture this variability.

Section 3.2 contains many novel and scientifically interesting results, in particular on the role of chlorophyll and circulation in driving variability in migration depth, and makes for a stimulating read. I like how the effects of different drivers are summarized in Fig. 7 and 8.

Line 432: you may be missing "studies" after "prior", or something analogous.

**Answer:** We thank the reviewer for pointing out this typo. We have corrected it ("prior" > "prior studies").

Line 455: this is an interesting point about diapause; I imagine that your ecological model could be in fact used in future work to explore tradeoffs between diapause vs. wintertime activity.

**Answer:** COBALTv2-DVM does not yet reproduce the mechanisms controlling diapause, which still appear to be uncertain, and exploring the trade-offs between diapause and wintertime should require further development. However, COBALTv2-DVM could already be used to explore trade-offs between migratory and non-migratory behavior, and their consequences for zooplankton growth, biogeochemical and trophic fluxes.

Figures are excellent. The color schemes of most figures are appropriate, but some use "jet" like palettes that are not optimal for visually impaired readers; I suggest the Authors revise them to adopt perceptually uniform palettes.

**Answer:** We thank the reviewer for this suggestion. We have replaced the panels of the figures in question (Fig 1a,b, 7a and Fig A4) to include perceptually uniform palettes.

**Figure 1:**

[Figure]

**Figure 7:**

[Figure]

Figure A4:

[Figure]